# Inhibitory circuit motifs of cortical somatosensory layer 5 SST interneurons are uniform within layers but specific across layers

Felix Preuss, Florian Walker [ID], Martin Möck, Mirko Witte and Jochen F. Staiger [ID]

*Department of Neuroanatomy, University Medical Center Göttingen, Göttingen, Germany*

Handling Editors: Katalin Toth & Conny Kopp-Scheinpflug

The peer review history is available in the Supporting Information section of this article (https://doi.org/10.1113/JP288309#support-information-section).

**Abstract figure legend** In our study we wanted to discover inhibitory circuits targeting layer 5 (L5) somatostatin (SST)-expressing neurons. As putative presynaptic cells we chose parvalbumin (PV) and vasoactive intestinal polypeptide (VIP)-expressing interneurons (IN). Previous studies showed that in L2/3 SST IN are targeted by PV and VIP cells. Here we tested intralaminar (L5 to L5) and translaminar (L2/3 to L5) connections. We discovered that L5 SST IN are targeted by L5 PV, L5 VIP and L2/3 VIP IN but not by L2/3 PV IN. Unitary synaptic properties and synaptic short-term plasticity displayed cell type-specific differences. L5 PV to L5 SST connections displayed larger amplitudes and shorter latencies than both VIP to L5 SST motifs. In addition we observed differences in synaptic short-term plasticity. L5 PV to L5 SST inputs were depressing at all tested frequencies (1, 8 and 40 Hz), whereas VIP inputs were facilitating but only at high-frequency 40 Hz stimulation. Our study helps to better understand dis/inhibitory processing in the cortical column.

**Abstract** Layer (L) 5 is a hub in the cortical column in which a multitude of feedforward and feedback pathways converge. These inputs are then transmitted to distant sites by resident pyramidal neurons (PN). L5 PN are under the strong influence of local somatostatin (SST)-expressing interneurons (IN). To better understand the inhibitory control of L5 SST cells, which leads to disinhibition of excitatory cells, we used paired whole-cell patch-clamp recordings in acute brain slices. We investigated whether they receive intra- and translaminar inputs by parvalbumin (PV) and vasoactive intestinal polypeptide (VIP) IN and what type of short-term synaptic plasticity these inputs display. In triple transgenic mice we found that intralaminarly both PV and VIP IN effectively target L5 SST IN. PV to SST connections were depressing at all tested frequencies, whereas VIP to SST connections were facilitating at high-frequency VIP IN stimulation. In addition translaminar inputs from L2/3 VIP to L5 SST IN showed similar connectivity rates and short-term plasticity compared to their L5 counterparts. However L2/3 PV IN, despite numerous descending axon collaterals, showed hardly any connection. In summary we are able to show that intralaminar circuit motifs of L5 SST IN resemble those previously studied in L2/3. Furthermore we demonstrate a selective translaminar targeting by L2/3 VIP IN that was missing from PV IN. These results shine new light on the circuit layout that enables intra- and translaminar dis/inhibitory processing in the cortical column.

(Received 13 December 2024; accepted after revision 31 October 2025; first published online 15 November 2025)

**Corresponding author** J. F. Staiger: Department of Neuroanatomy, University Medical Centre Göttingen, Göttingen, Germany. Email: jochen.staiger@med.uni-goettingen.de

**Key points**

- L5 somatostatin (SST)-expressing interneurons (IN) are widely targeted by other GABAergic IN in their home layer.
- The origin of this afferent inhibition is from parvalbumin (PV), as well as vasoactive intestinal peptide (VIP)-expressing GABAergic IN within L5, whereas L2/3 translaminar input originates from VIP but not from PV IN.
- Many of these connections are formed in a bidirectional manner.
- PV to SST and VIP to SST connections displayed cell type-specific differences in unitary synaptic properties and short-term plasticity.
- Our results help us to better understand intra- and translaminar dis/inhibitory processing, which might optimize tactile information processing in the cortical column.

## Introduction

Facial whiskers represent a core sensory organ of rodents to actively gather information of their immediate surroundings. This tactile information is processed in the whisker region of the primary somatosensory cortex (wS1), also called barrel cortex (Woolsey & Van der Loos, 1970). Like most other neocortical regions wS1 consists of multiple layers, which are considered to perform important functions in sensory information processing (Guy & Staiger, 2017; Harris & Mrsic-Flogel, 2013; Harris & Shepherd, 2015; Miller et al., 2001; Schubert et al., 2007; Staiger & Petersen, 2021). However

**Felix Preuss** completed his master's degree in medical biology at the University of Duisburg-Essen and received his PhD from the Göttingen Graduate Centre for Neurosciences, Biophysics and Molecular Biosciences (GGNB) while working at the Department of Neuroanatomy at the University Medical Centre Göttingen in 2024. He is currently working as a postdoctoral researcher at the Neurophysics Laboratory at the Georg-August University Göttingen. Dr. Preuss is particularly interested in inhibitory interneurons and their circuitry in the mammalian neocortex.

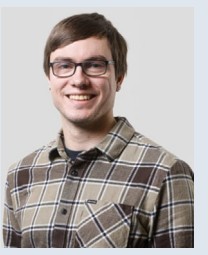

precise layer-specific processing mechanisms supported by experimental evidence are only slowly appearing (Adesnik & Naka, 2018; Brecht, 2007; Staiger & Petersen, 2021; Varani et al., 2022). In addition to intralaminar signal processing a hallmark of cortical circuitry is a multitude of translaminar circuits, involving excitatory as well as inhibitory neurons whose structural layout and functional implications are even less understood (Campagnola et al., 2022; Feldmeyer, 2012; Naka et al., 2019; Staiger & Petersen, 2021).

A previous consensus was that GABAergic interneurons (IN) can be subdivided into different groups based on the expression of different key molecular markers (Rudy et al., 2011; Staiger et al., 2015; Xu et al., 2010). Most recent transcriptomic analysis suggests four different groups: parvalbumin (PV), somatostatin (SST), vasoactive intestinal peptide (VIP) and other marker (Lamp5-, Sncg- and Serpinf-1)-expressing cells, which can be further subdivided into multiple cell types or even subtypes (Mao & Staiger, 2024; Tasic et al., 2018). These subtypes differ in morphology, electrophysiological properties, expression of other molecular markers, local- and long-range inputs, target specificity and synapse location (Callaway et al., 2021; Feldmeyer et al., 2018; Staiger & Petersen, 2021). Altogether the interplay of excitatory and different subtypes of inhibitory neurons enables a dynamic excitation-inhibition ratio tailored to context-dependent neural processing in the neocortex (Haider et al., 2006; Wilent & Contreras, 2005; Xue et al., 2014; Yizhar et al., 2011).

IN not only control excitatory cells but also target other IN, a circuit motif which can lead to disinhibition of principal cells (Feldmeyer et al., 2018; Pfeffer et al., 2013). The best-studied circuit motif of this kind is the VIP to SST motif, which has been described in multiple cortical regions (Fu et al., 2014; Kullander & Topolnik, 2021; Lee et al., 2013; Pi et al., 2013; Walker et al., 2016). The most prominent morphological subpopulations of SST IN are the Martinotti cells (MC), which are characterized by extensive axonal arborization in L1, where they are thought to target the apical tufts of pyramidal neurons (PN) (Ma et al., 2006; McGarry et al., 2010; Wang et al., 2004). In addition there are SST IN lacking the L1-reaching axonal component, the so-called non-Martinotti cells (nMC) (Ma et al., 2006). VIP IN targeting SST IN are effectively reducing the inhibitory drive onto PN (Fu et al., 2014). Surprisingly widespread connections of SST IN with other IN have been described recently (Donato et al., 2023). Moreover also PV IN have been shown to act as 'disinhibitors', as they target not only other PV but also SST IN in multiple cortical areas (Campagnola et al., 2022; Jiang et al., 2015; Walker et al., 2016).

L5 is known as the output layer of the neocortex, but it also possesses rich input connectivity, with number of feedback and feedforward pathways converging there (Feldmeyer, 2012; Feldmeyer et al., 2018; Kampa et al., 2006; Schubert et al., 2006). In addition a substantial fraction of SST IN can be found in L5, which are much more diverse than their supragranular counterparts (Gouwens et al., 2019; Halabisky et al., 2006; Ma et al., 2006; Nigro et al., 2018; Tremblay et al., 2016; Wu et al., 2023). MC in L5 can be subdivided into two different morphological subclasses namely the T-shaped and fanning out MC, named after their distinctive axonal arborization (Muñoz et al., 2017; Nigro et al., 2018). In addition to MC ≈35% of L5 SST IN can be morphologically classified as nMC. Both MC subclasses and nMC in L5 differ in morphology, electrophysiological properties and target specificity (Nigro et al., 2018). L5 MC target L5 PN, and L5 nMC target L4 spiny stellate neurons, which constitute the main excitatory cell population in L4 of the wS1. In addition SST subclasses are active during different states of behaviour (Muñoz et al., 2017). Altogether these findings suggest that L5 SST IN play important roles in sensory processing. Most recent studies have focused on the connectivity of V1 (Campagnola et al., 2022; Jiang et al., 2015); therefore little is known about how L5 SST IN in wS1 are controlled by other IN.

Here we used optogenetics and paired patch-clamp recordings to study inhibitory inputs to L5 SST IN. To further extend our knowledge of these circuit motifs we not only recorded from putative presynaptic PV and VIP IN in L5 but also tested for translaminar connections established by L2/3 PV and VIP IN. We analysed basic synaptic properties and short-term plasticity at different stimulation frequencies in PV to SST and VIP to SST IN connections. We found that L5 PV, L5 VIP and L2/3 VIP, but not L2/3 PV IN, reliably target L5 SST IN with cell type-specific differences in basic synaptic properties and short-term synaptic plasticity. Taken together we were able to expand our knowledge of inhibitory circuitry in mouse wS1, further strengthen the additional role of PV IN as local 'disinhibitors' and the existence of the VIP to SST motif as an intra- and translaminar disinhibitory circuit motif, a connectivity more frequently organized in a bidirectional manner than previously assumed.

## Results

We used two different triple-transgenic mouse lines (PV-Cre//tdTomato//GIN and VIP-Cre//tdTomato//GIN) to investigate disinhibitory circuit motifs of L5 SST neurons in mouse wS1 (Fig. 1A). Cre-driver lines are an essential tool to investigate neuronal circuits and have been successfully used over a decade (Taniguchi et al., 2011). In the GIN mouse a subpopulation of SST IN expresses the green fluorescent protein GFP (Oliva et al., 2000; Zhou et al., 2020). In previous studies it has been shown that

in triple-transgenic animals PV, VIP and SST IN retain cell type-specific morphological and electrophysiological characteristics (Walker et al., 2016).

Although the majority of SST IN in L2/3 can be described as MC (Fanselow et al., 2008; Hostetler et al., 2023; Ma et al., 2006; McGarry et al., 2010), the population of L5 SST IN is much more diverse in terms of electrophysiological properties and morphology (Nigro et al., 2018). We analysed 238 SST IN in L5 that were recorded with a caesium-based internal solution. Of those 84.0% (200/238) were located in L5a, and 16.0% (38/238) were located in L5b. Analysis included morphological classification as Martinotti (MC, see Fig. 1*B*) and non-Martinotti cells (nMC, see Fig. 2*E*), as well as electro-physiological characterization based on input resistance at steady state ($R_i$), membrane time constant ($\tau$) and sag index. We were able to morphologically classify 155 cells of which 124 (80.0%) were MC and 31 (20.0%) were nMC (Fig. 2*A*); 83 of 238 cells were either not recovered at all or only partially; thus we were not able to perform an unambiguous classification. Identified MCs had significantly lager $R_i$ (MC: 264.7 ± 84.75 MΩ; $n$ = 124; nMC: 161.8 ± 67.06 MΩ, $n$ = 31; $P$ < 0.0001, Mann–Whitney test, Fig. 2*B*), $\tau$ (MC: 29.82 ± 9.18 ms, $n$ = 124; nMC: 16.04 ± 6.12 ms, $n$ = 31; $P$ < 0.0001, Mann–Whitney test, Fig. 2*C*) and sag indices (MC: 7.49 ± 4.25%, $n$ = 123; nMC: 4.68 ± 3.20%, $n$ = 30; $P$ = 0.0006, Mann–Whitney test, Fig. 2*D*). Nevertheless

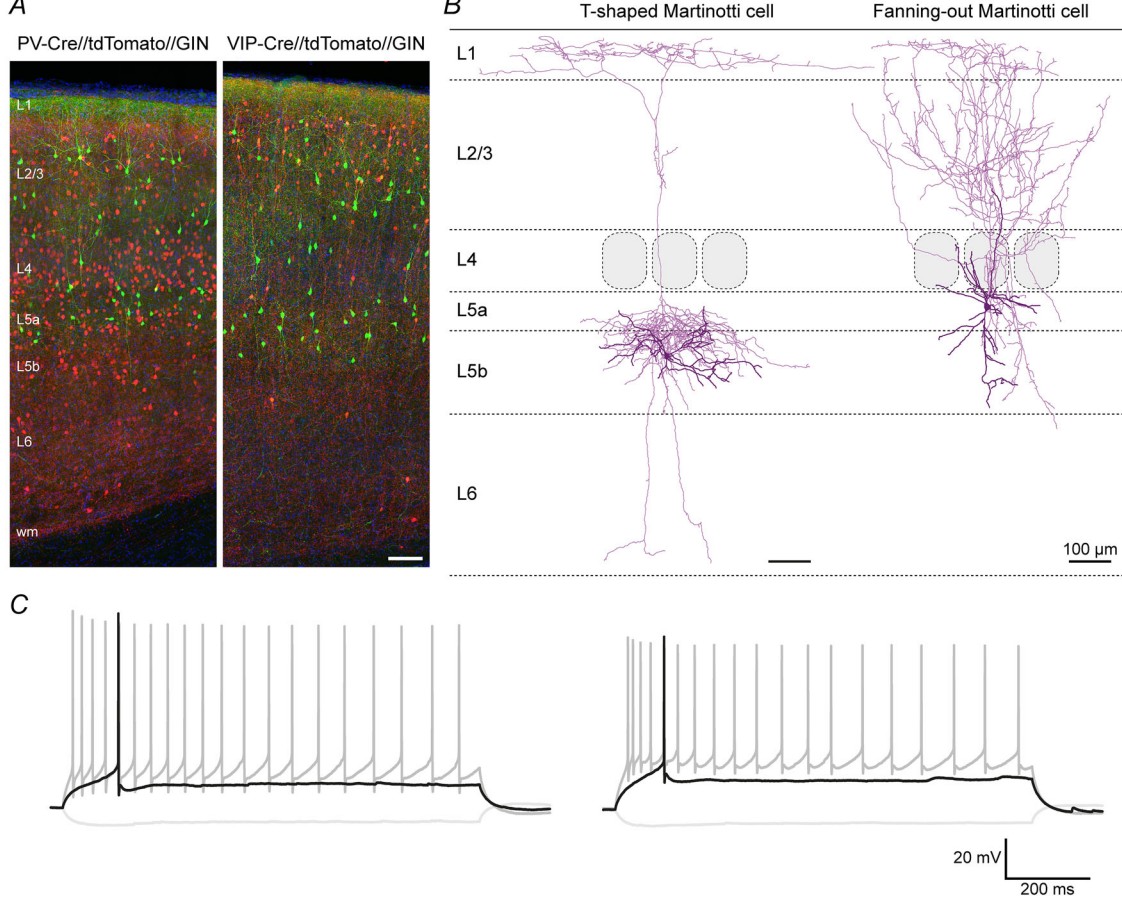

**Figure 1. L5 GIN cells display somatostatin (SST) interneuron (IN) typical morphological and electro-physiological properties**

*A*, maximum-intensity projections of confocal images of a 300 µm-thick thalamocortical slice of a parvalbumin (PV)-Cre//tdTomato//GIN (left) and a vasoactive intestinal polypeptide (VIP)-Cre//tdTomato//GIN (right) animal from wS1. PV or VIP IN are shown in red and GIN cells in green. In the GIN mouse line the majority of SST IN are located in L2/3. Infragranular SST IN are predominantly located in L5a. *B*, morphological reconstructions of a T-shaped Martinotti cells (MC) (L5b, left) and a fanning-out MC (L5a, right). SST soma and dendrites are shown in dark and axon in light purple. Both MC IN show the typical L1-reaching axonal component of MC together with strong ramification within L1. Scale bars = 100 µm. *C*, exemplar recordings of SST IN at different rectangular current injections recorded with a potassium gluconate-based internal solution. A hyperpolarizing step (−50 pA; light grey), a single action potential at rheobase (60 pA left, 70 pA right; black) and the typical continuous adapting firing pattern of SST IN are shown (120 pA left, 130 pA right; medium grey).

there was a strong overlap between the group of MC and nMC, which did not allow us to perform a cell type classification solely based on the electrophysiological data in individual cells. We therefore decided to not further subdivide our datasets into morphologically identified subtypes for statistical analysis.

### L5 SST IN receive inhibitory inputs from PV and VIP IN

In previous studies PV and VIP IN were shown to locally inhibit SST IN in L2/3 of mouse wS1 (Walker et al., 2016). We wanted to test whether this disinhibitory circuit motif is layer-specific or if it can also be found in L5, which is, based on the canonical microcircuit, the main output layer of the large whisker representation in wS1. Therefore we expressed ChR2 in PV and VIP IN by injecting the Cre-dependent virus AAV6-EF1a-double floxed-hChR2(H134R)-mCherry-WPRE-HGHpA into mouse barrel cortex of either PV-Cre or VIP-Cre animals. ChR2 was expressed throughout the whole cortical depth, and we recorded from L5 SST IN. Light activation of ChR2 (100 μm diameter spot with SST cell soma in the centre) reliably evoked IPSCs in L5 SST IN (PV: 18/19 (94.7%); VIP: 14/14 (100%) Fig. 3C), suggesting a strong

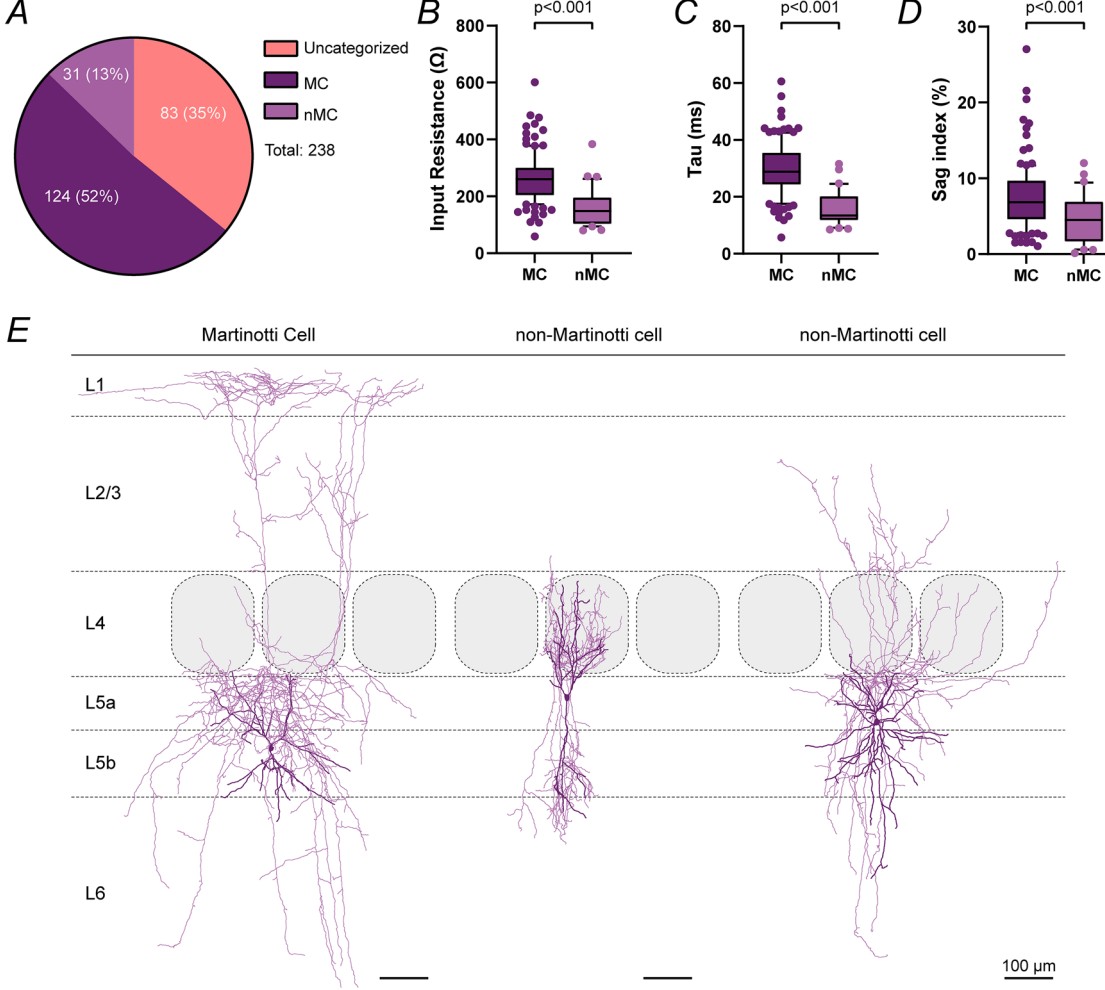

**Figure 2. The majority of L5 SST IN in the GIN mouse can be classified as Martinotti cells (MC) and display significantly different passive membrane properties in comparison to non-MC (nMC)**

*A*, pie chart of 238 recorded L5 SST IN. Neurons characterized as MC are shown in dark purple (124 (≈52%)) and as nMC in light purple (31 (≈13%)). IN that did not recover, where the axon was truncated by slicing, or where we were not able to make a safe classification for other reasons are shown as 'uncategorized' in light red (83 (≈35%)). *B–D*, quantitative analysis of passive properties of all 155 morphological identified SST IN from A (MC: *n* = 124, nMC: *n* = 31). The following parameters have been analysed: input resistance (*B*), membrane time constant (*C*) and sag index (*D*). MC and nMC were significantly different in all three parameters. Data are shown as 10%–90% box and whisker plots. *E*, morphological reconstructions of one MC (extensively innervating L1) and two nMC lacking this L1 innervation, which is not the effect of slicing artefacts. Soma and dendrites are shown in dark purple and axon in light purple. Scale bar = 100 μm.

targeting of L5 SST by PV and VIP cells, comparable to L2/3 SST cells. Laser threshold to induce IPSCs was 101.5 ± 92.4 µW for PV ($n = 11$) and 192.8 ±163.8 µW for VIP IN ($n = 13$, $P = 0.0904$, Mann–Whitney test, Fig. 3*D*). IPSC at 10× laser threshold amplitude

was 352.7 ± 252.4 pA for PV inputs ($n = 11$) and 254.2 ± 240.0 pA for VIP inputs ($n = 13$, $P = 0.3311$, Mann–Whitney test, Fig. 3*E*). These findings demonstrate a strong functional innervation of L5 SST IN from PV and VIP IN.

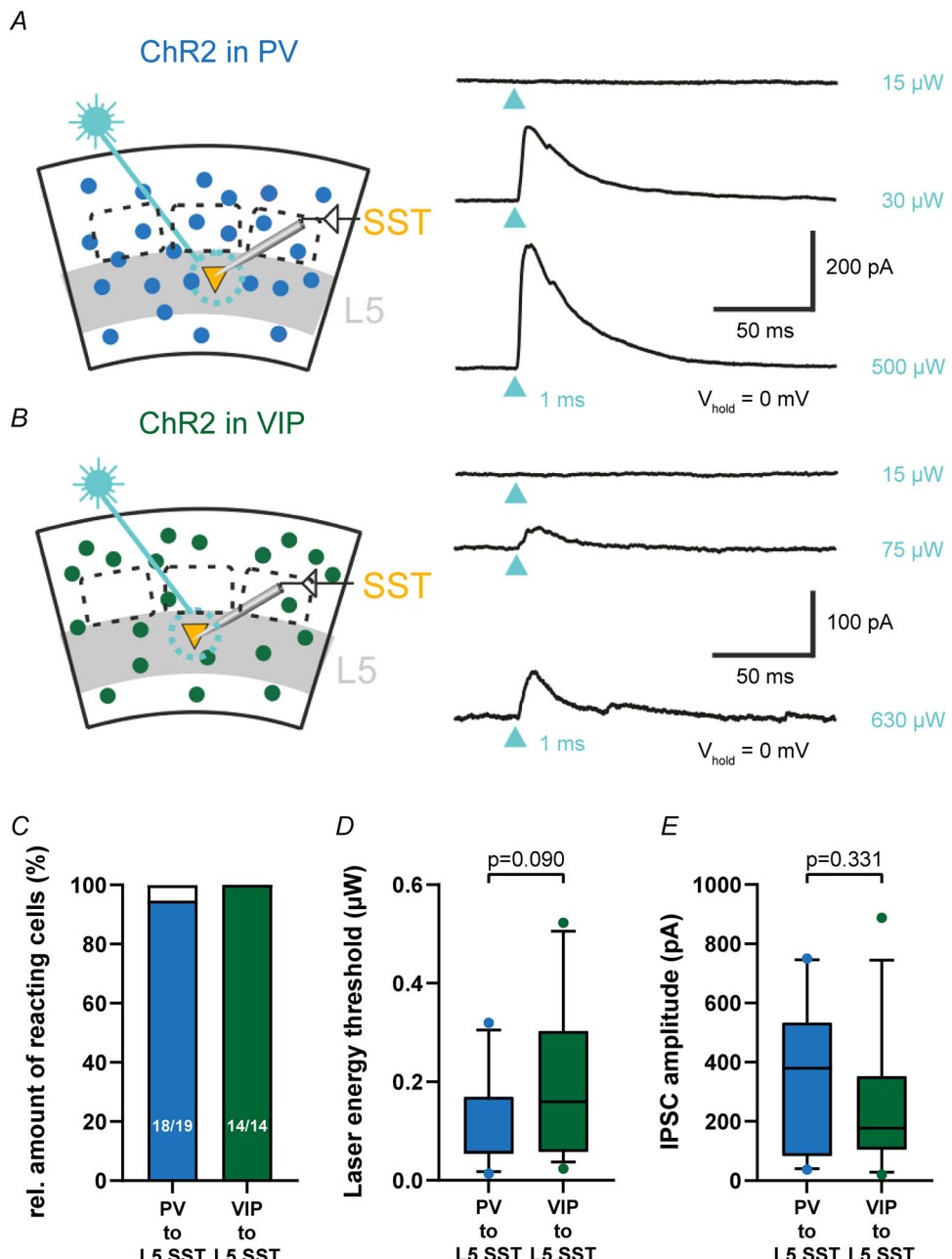

**Figure 3. At a population level parvalbumin (PV) and vasoactive intestinal polypeptide (VIP) inter-neurons (IN) reliably target layer 5 somatostatin (L5 SST) IN**

*A, B*, left: schematic recording configuration. L5 SST IN were recorded during photostimulation of ChR2-expressing PV (*A*) or VIP IN (*B*). Right: examples of photostimulation-induced IPSCs. Light blue arrowheads indicate timing of photostimulation (1 ms, 473 nm laser) of PV (*A*) and VIP IN (*B*) at three different intensities (subthreshold, threshold and multitude of threshold). *C–E*, quantification of PV and VIP IN photostimulation on postsynaptic L5 SST IN. *C*, fraction of responsive L5 SST IN. Box plots (PV to L5 SST: $n = 11$, VIP to L5 SST: $n = 13$) of laser energy threshold (*D*) and IPSC amplitude at 10× laser threshold *E*. Overall there was no significant difference between PV and VIP IN, suggesting a strong functional innervation of L5 SST IN by PV and VIP IN.

Considering that we photostimulated the area around the soma of L5 SST IN we likely activated the axons of a population of both local and distant PV and VIP IN. Especially L2/3 VIP IN display dense axonal arborization in L5 (Prönneke et al., 2015). Therefore we hypothesized that L5 SST IN may not only be targeted intralaminarly by local but also translaminarly by L2/3 VIP IN. Furthermore axon collaterals of L2/3 PV IN also do frequently descend into infragranular layers (Gouwens et al., 2019; Jiang et al., 2015). However this morphological feature is not as common and rich as in L2/3 VIP IN (Gouwens et al., 2019; Prönneke et al, 2015), which does, however, not preclude a specific targeting of L5 SST IN. Consequently we wanted to specifically test these intra- and translaminar circuit motifs by paired whole-cell recordings of layer-specified single presynaptic IN.

## Unitary synaptic properties

The present optogenetic experiments and previously known morphological features of L2/3 VIP and PV cells (Gouwens et al., 2019; Jiang et al., 2015; Prönneke et al., 2015) suggested that L5 SST IN could be targeted not only in an intralaminar but also in a translaminar way. Therefore we performed four types of paired recordings of potential presynaptic neurons: (i) L5 PV, (ii) L5 VIP, (iii) L2/3 PV or (iv) L2/3 VIP IN and postsynaptic L5 SST IN. Exemplar recordings and morphologies of individual IN pairs that are shown in Fig. 4 already illustrate some of the core reproducible features: (i) in terms of unitary connections, despite all fluctuations in individual traces, the averaged amplitude is much larger in the L5 PV to SST connections than in both the L5 VIP and the L2/3 VIP to SST IN connection (Fig. 4*A* and *B*). (ii) In terms

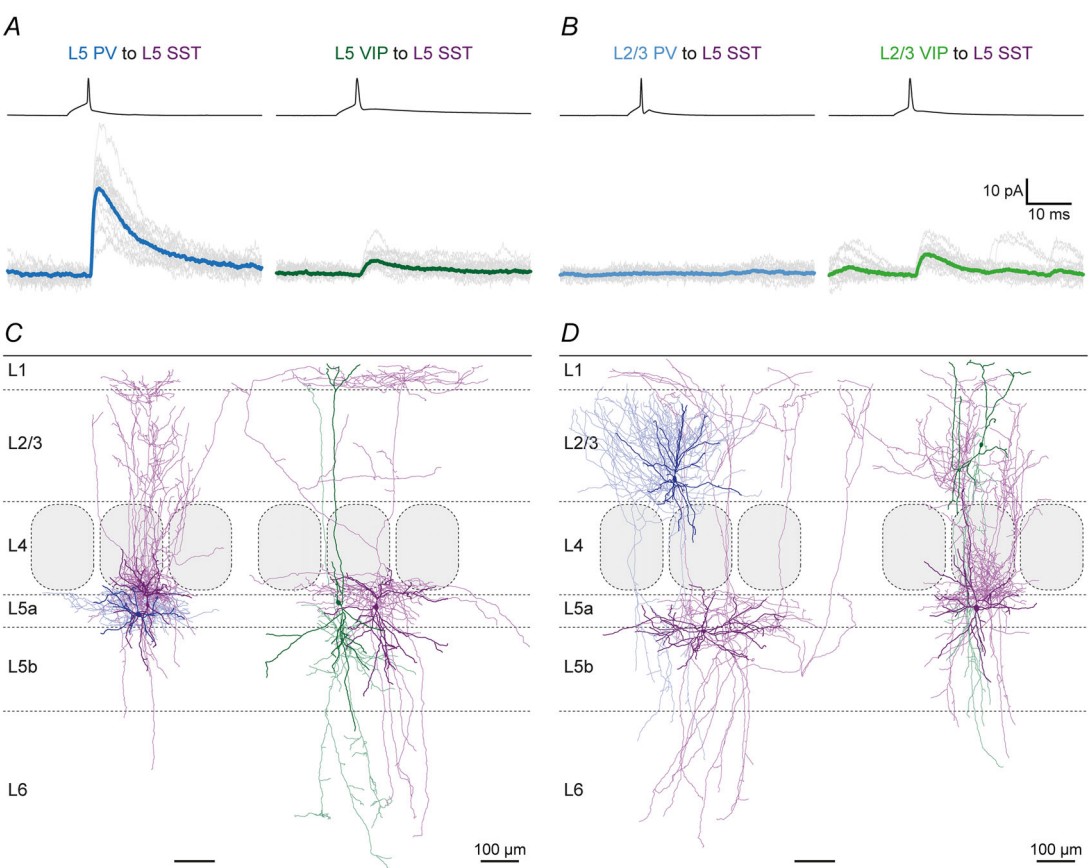

**Figure 4. Individual parvalbumin (PV) and vasoactive intestinal polypeptide (VIP) interneurons (IN) locally target L5 SST IN, whereas translaminar connections were only found in L2/3 VIP to L5 SST IN pairs**

Electrophysiological recordings of intralaminar (*A*) and translaminar (*B*) PV and VIP to L5 SST IN pairs. SST IN were held at 0 mV to promote recording of unitary IPSCs. Individual responses of strikingly different amplitude size are shown in grey and the averages in blue or green. No connection was found in the exemplar translaminar L2/3 PV to L5 SST IN pair. Morphological reconstructions of intralaminar (*C*) and translaminar (*D*) PV and VIP to L5 SST IN recordings. SST IN soma and dendrites are shown in dark and axon in light purple; PV IN soma and dendrites are shown in dark and axon in light blue; VIP IN soma and dendrites are shown in dark and axon in light green. All L5 SST IN resemble Martinotti cells (MC), whereas PV IN can be considered as basket cells and VIP IN as bipolar-bitufted cells with descending axon. Scale bar = 100 μm.

of morphology the presynaptic L5 PV IN displayed a clear basket cell (BC) morphology, in contrast to the L5 VIP IN that exhibited a vertically extended dendritic tree and a strictly infragranular axonal arbor, whereas the post-synaptic L5 SST IN were classified as MC (Fig. 4*C*); in Fig. 4*D*, again, the L5 SST IN represented MC, whereas the L2/3 PV IN was a BC with descending axonal collaterals, and the L2/3 VIP IN was a bipolar neurons with a strongly arborizing descending axon. In the additional galleries of all reconstructed connected IN pairs (Figs. 9–11), by separately showing the pre- and the postsynaptic neuron, the details of the single-cell morphologies become much clearer. It can be appreciated that L5 PV IN exhibited a typical BC morphology with largely local axonal arbors (Fig. 9), whereas VIP IN were stretched out radially across multiple layers with both their dendrites and axons (Figs. 10 and 11). L5 SST IN could be separated into MC and nMC forms, depending on whether they innervated L1 with their ascending axons (Figs. 9–11).

To optimize chances of retained connectivity we usually stayed below 200 μm soma distance in intralaminar IN pairs. For translaminar IN pairs we aimed at patching in the same cortical column. We found comparable connectivity rates in intralaminar IN pairs between L5 PV to L5 SST (24.7%; 18/73) and L5 VIP to L5 SST (21.1%; 15/71) pairs. In the translaminar L2/3 VIP to L5 SST connection we observed the highest connectivity rate with 29.6% (16/54). Although we frequently observed L2/3 PV IN with descending axon collaterals (e.g. Fig. 4*D*), we only discovered a single connected L2/3 PV to L5 SST pair in 48 trials (2.8%; 1/48, Fig. 6*D*). Because these cells were frequently connected in the reverse direction (see below), we considered the L2/3 PV to L5 SST connection very rare and not a technical artefact. Reverse (L5 SST to PV/VIP IN, Fig. 5*C*) connectivity rates were high throughout: L5 SST to L5 PV: 35.5% (11/31); L5 SST to L5 VIP: 23.1% (9/39); L5 SST to L2/3 PV: 36.6% (15/41); L5 SST to L2/3 VIP: 23.9% (11/46). Unfortunately the caesium-based intracellular solution used in L5 SST IN massively changes the parameters of action potentials (Fig. 5*A*). Therefore we did not further analyse the elementary properties of reversed connections and only report their connectivity rate.

We were able to analyse evoked unitary responses in 13 L5 PV to L5 SST IN pairs, 11 L5 VIP to L5 SST IN pairs and 13 L2/3 VIP to L5 SST IN pairs (parameters of analysed pairs are shown in Table 1).

Presynaptic PV action potentials resulted in reliable postsynaptic IPSCs (synaptic success rate L5 PV to L5 SST: $87.61 \pm 13.91\%$, $n = 13$), whereas both VIP connections were much less reliable (synaptic success rate: L5 VIP to L5 SST: $52.04 \pm 18.81\%$, $n = 11$; L2/3 VIP to L5 SST: $54.96 \pm 23.57\%$, $n = 13$, Fig. 6*D*). Analysing the IPSC parameters we found significant differences between the

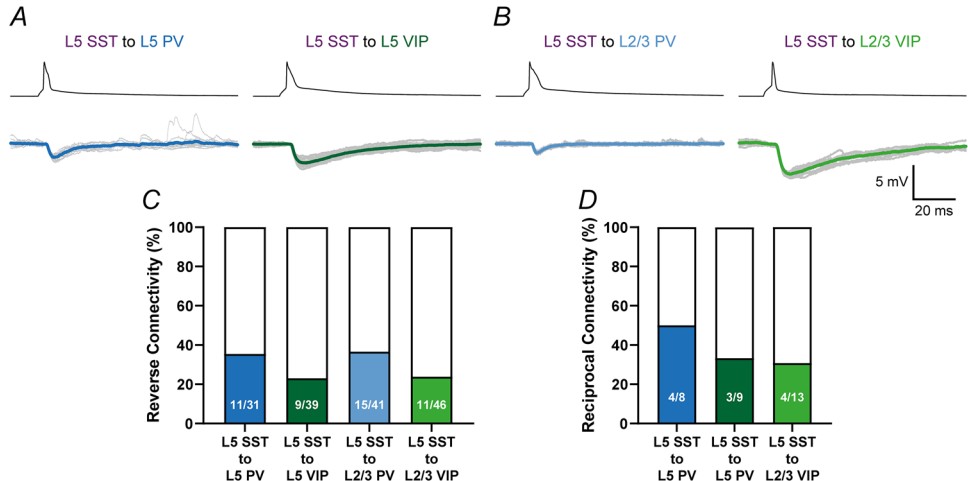

**Figure 5. Layer 5 somatostatin (L5 SST) interneurons (IN) effectively target parvalbumin (PV) and vaso-active intestinal polypeptide (VIP) IN in an intralaminar and translaminar way**
Electrophysiological recordings of intralaminar (*A*) and translaminar (*B*) 'reverse' SST to PV or SST to VIP connections. Postsynaptic PV and VIP IN were depolarized and held close to their respective firing threshold (ca −48 mV) in current clamp. Individual responses are shown in grey and averages in blue or green. Action potentials of pre-synaptic SST IN (black) are broadened due to the use of a caesium-based intracellular solution. *C*, connectivity rates of all four tested reverse connections (SST to PV or SST to VIP). Intra- and translaminar SST to PV IN pairs had similar connectivity rates (L5 SST to L5 PV: 35.5%; L5 SST to L5 SST to L2/3 PV: 36.6%) and were higher than both SST to VIP IN connections (L5 SST to L5 VIP: 23.1%; L5 SST to L2/3 VIP: 23.9%). *D*, probability to find reciprocally connected pairs. After experiments in some pairs we also checked for the reciprocal connection. We found reciprocal connections in 50% (4/8) of the tested L5 PV to L5 SST IN pairs, in 33.3% (3/9) of the tested L5 SST to L5 VIP IN pairs and in 30.7% (4/13) of the tested L5 SST to L2/3 VIP IN pairs. The only L2/3 PV to L5 SST IN pair that we found was reciprocally connected as well.

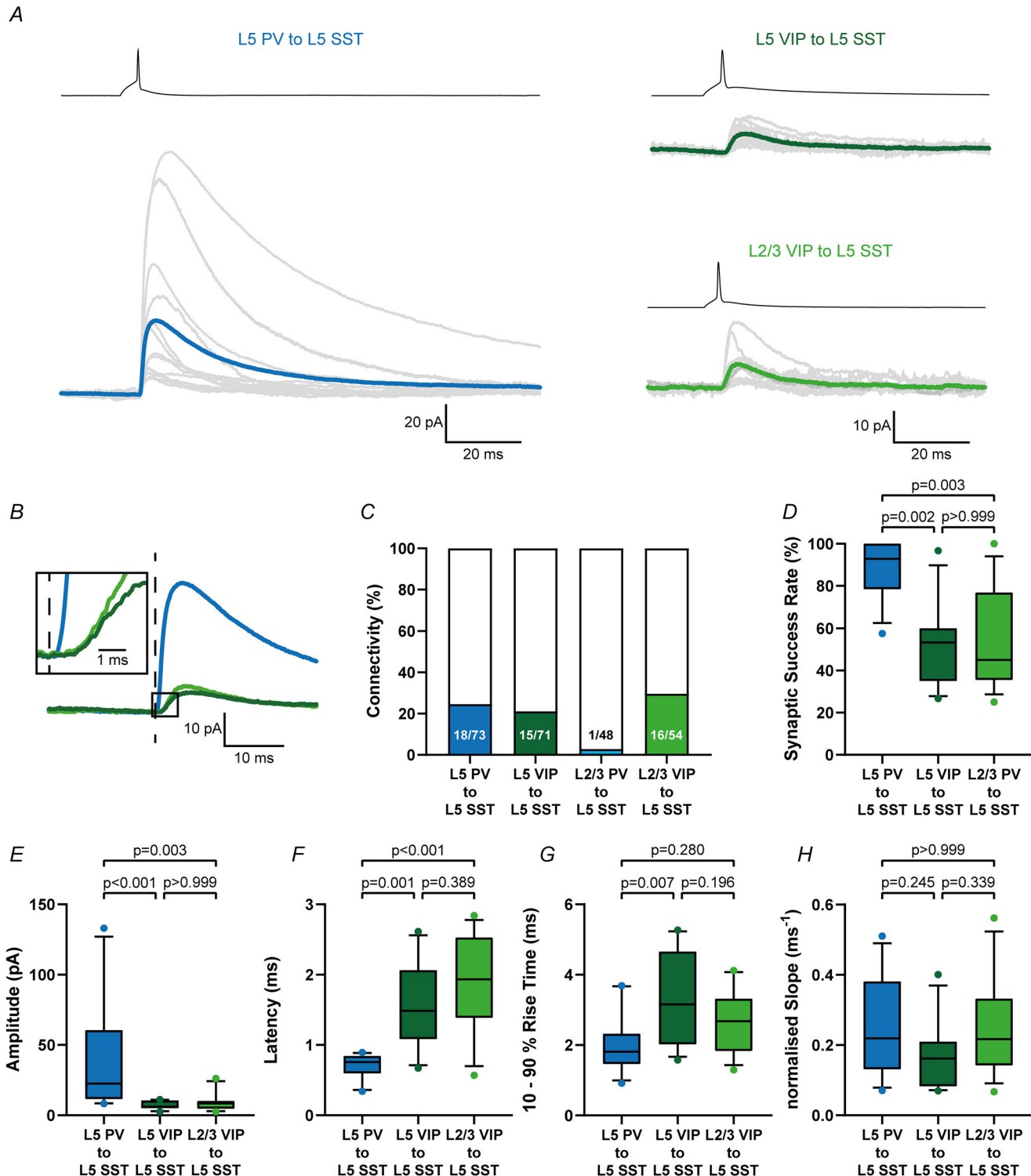

**Figure 6. Elementary properties of intra- and translaminar synaptic inputs of parvalbumin (PV) and vasoactive intestinal polypeptide (VIP) interneuron (IN) to layer 5 (L5) somatostatin (SST) IN possess presynaptic IN type-specific differences**

*A*, grand averages of unitary IPSCs for all analysed connections (L5 PV to L5 SST: *n* = 13; L5 VIP to L5 SST: *n* = 11; L2/3 VIP to L5 SST: *n* = 13) in postsynaptic SST IN in response to evoked single spikes in presynaptic IN. Averages of individual connections are shown in grey and the grand averages in blue (L5 PV to L5 SST), dark green (L5 VIP to L5 SST) and light green (L2/3 VIP to L5 SST). *B*, overlay of all three analysed grand averages from (*A*) aligned to presynaptic spike peaks. PV- and VIP-evoked IPSCs differ substantially in amplitude size and kinetics.

For better visualization the IPSC onset period is magnified in the rectangular box. The dashed line indicates the presynaptic spike peak. *C*, connectivity rate of all four tested connections. The highest connectivity rate was found in the translaminar L2/3 VIP to L5 SST connection with 29.6% (16/54). The probability to discover intralaminar connections was at 24.7% for L5 PV to L5 SST IN (18/73) and at 21.1% for L5 VIP to L5 SST IN (15/71). For the translaminar L2/3 PV to L5 SST connection we only found a single connected pair in 48 trials (2.8%). *D*, synaptic transmission of PV IN was more reliable than in both VIP connections. *E–H*, quantification of unitary IPSCs. The parameters amplitude (*E*), latency (*F*), 10%–90% rise time (*G*) and normalized slope as amplitude divided by time (*H*) have been analysed on averages of individual pairs (L5 PV to L5 SST: blue, *n* = 13; L5 VIP to L5 SST: dark green, *n* = 11; L2/3 VIP to L5 SST: light green, *n* = 13. Data are shown as 10%–90% box-and-whisker plots.

L5 PV to L5 SST and both VIP connections (Fig. 6*E* and *F*), whereas intra - *versus* translaminar VIP connections did not show any significant differences (*P*-values are shown in Table 2).

Compared to intra- and translaminar VIP to SST IN connections the intralaminar PV to SST IN connection was larger in amplitude (L5 PV to L5 SST: 42.18 ± 41.52 pA, *n* = 13; L5 VIP to L5 SST: 7.03 ± 2.92 pA, *n* = 11; L2/3 VIP to L5 SST: 9.63 ± 6.85 pA, *n* = 11), shorter in latency (L5 PV to L5 SST: 0.70 ± 0.18 ms, *n* = 13; L5 VIP to L5 SST: 1.58 ± 0.62 ms, *n* = 11; L2/3 VIP to L5 SST: 1.88 ± 0.69 ms, *n* = 13) and had a faster 10%–90% rise time (L5 PV to L5 SST: 1.99 ± 0.86 ms, *n* = 13; L5 VIP to L5 SST: 3.33 ± 1.26 ms, *n* = 11; L2/3 VIP to L5 SST: 2.60 ± 0.91 ms, *n* = 13). These differences in amplitude and kinetics

can easily be identified by visual comparison of the grand averages of each connection (Fig. 6*B*). Only for the normalized slope we did not see any significant differences between the three connections (L5 PV to L5 SST: 0.26 ± 0.14 fraction of amplitude ms$^{-1}$, *n* = 13; L5 VIP to L5 SST: 0.17 ± 0.10 fraction of amplitude ms$^{-1}$, *n* = 11; L2/3 VIP to L5 SST: 0.25 ± 0.14 fraction of amplitude ms$^{-1}$, *n* = 13). In all three connections we also found bidirectionally connected pairs (Fig. 5*D*). In the L5 PV to L5 SST connection 4 out of 8 (50.0%) tested pairs were reciprocally connected, which was the highest rate. In the intralaminar L5 VIP to L5 SST connection 3 out of 9 (33.3%), and in the translaminar L2/3 VIP to L5 SST connection 4 out of 13 (30.8%) tested pairs were bidirectionally connected. These and previous findings from L2/3 suggest that the PV to SST and VIP to SST

**Table 1. Parameters of analysed IN pairs**

| | *n* | Animals | Minimum age | Median age | Maximum age | Fraction of SST IN in L5a |
|---|---|---|---|---|---|---|
| L5 PV to L5 SST | 13 | 11 | P22 | P32 | P44 | 8/13 (61.5%) |
| L5 VIP to L5 SST | 11 | 11 | P27 | P33 | P48 | 9/11 (81.8%) |
| L2/3 VIP to L5 SST | 13 | 12 | P24 | P43 | P53 | 9/13 (69.2%) |

*Notes*: Table containing the number of analysed pairs (*n*), number of animals, minimum, median and maximum age of used animals (postnatal days) and the fraction of cells that were located in L5a of analysed IN pairs. In general there were no large differences between those parameters. In the GIN mouse line the majority of infragranular SST IN are located in L5a. Therefore the majority of patched SST IN are located in L5a.
Abbreviations: IN, interneurons; L5, layer 5; PV, parvalbumin; SST, somatostatin; VIP, vasoactive intestinal polypeptide.

**Table 2. *P*-values of unitary synaptic properties parameters**

| | Amplitude | Latency | 10%–90% rise time | Normalised slope | Synaptic success rate |
|---|---|---|---|---|---|
| Test | Kruskal–Wallis test | One-way ANOVA | One-way ANOVA | Kruskal–Wallis test | Kruskal–Wallis test |
| L5 PV to L5 SST *vs.* L5 VIP to L5 SST | 0.0009 | 0.0010 | 0.0071 | 0.2451 | 0.0017 |
| L5 PV to L5 SST *vs.* L2/3 VIP to L5 SST | 0.0033 | <0.0001 | 0.2800 | >0.9999 | 0.0028 |
| L5 VIP to L5 SST *vs.* L2/3 VIP to L5 SST | >0.9999 | 0.3890 | 0.1955 | 0.3390 | >0.9999 |

*Notes*: Table containing the *P*-values and used statistical test for the analysis of amplitude, latency, 10%–90% rise time, normalized slope and synaptic success rate. If data were normally distributed, a one-way ANOVA was used. Otherwise the Kruskal–Wallis test was used. Most parameters were significantly different when comparing the PV to SST connection to either VIP to SST connection. There was no significant difference between both VIP to SST connections.
Abbreviations: L5, layer 5; PV, parvalbumin; SST, somatostatin; VIP, vasoactive intestinal polypeptide.

connectivity motifs are highly conserved throughout the laminar hierarchy of the cortex.

## Short-term synaptic plasticity

*In vivo* synaptic transmission usually occurs with trains of multiple presynaptic action potentials at varying frequencies, impinging on and being integrated by post-synaptic neurons (Gentet et al., 2010, 2012). Under such conditions synaptic connections are not static but display varying forms and degrees of short-term dynamics (Campagnola et al., 2022; Markram & Tsodyks, 1996; Markram et al., 1998; Stevens & Wang, 1995). In previous studies we observed different short-term plasticity in PV to SST and VIP to SST connections in L2/3 wS1 (Walker et al., 2016). To examine whether this is a specific feature of L2/3 connections or whether it can be generalized to other layers as well we analysed synaptic short-term plasticity of L5 SST IN input connections. We triggered trains of five action potentials at 1, 8 and 40 Hz frequency in presynaptic PV or VIP cells (Fig. 7). Also in L5 SST IN we saw IN type-specific differences in short-term plasticity. The L5 PV to L5 SST connection

was depressing at all tested frequencies (Fig. 8*A*, *D* and *G*). This effect became stronger with increased presynaptic firing frequency (normalized amplitude reduction in the fifth response: 1 Hz: 31.77 ± 15.24%, *n* = 9; 8 Hz: 45.31 ± 14.73%, *n* = 8; 40 Hz: 58.00 ± 13.06%, *n* = 8; mean ± SD).

In contrast both VIP connections exhibited prominent short-term facilitation at 40 Hz stimulation frequency (1 Hz: L5 VIP to L5 SST *n* = 11, L2/3 VIP to L5 SST *n* = 11; 8 Hz: L5 VIP to L5 SST *n* = 12, L2/3 VIP to L5 VIP *n* = 12; 40 Hz: L5 VIP to L5 SST *n* = 13, L2/3 VIP to L5 SST *n* = 13). At lower frequencies we mostly did not see an apparent change in amplitude size, giving the impression of a 'stable' connection. At 1 Hz stimulation we partially observed depression in the translaminar L2/3 VIP to L5 SST connection (Fig. 8*C*). However this effect was only observable in the 4th and 5th responses (normalized amplitude reduction in 4th response: 29.37 ± 30.52%; normalized amplitude reduction in the 5th response: 23.06 ± 29.23%; mean ± SD, *n* = 11). At 8 Hz (Fig. 8*E* and *H*) we did not observe plasticity at any evoked response in the intralaminar and translaminar VIP to SST connections, respectively. At high-frequency stimulation (40 Hz, Fig. 8*F* and *I*) both VIP connections

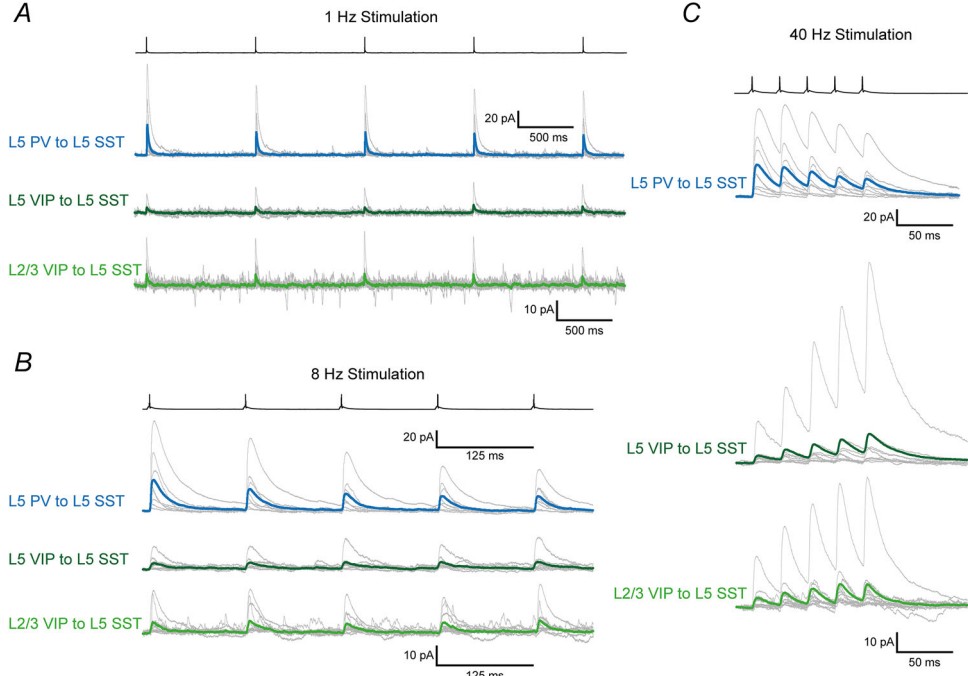

**Figure 7. Intralaminar parvalbumin (PV) and intra/translaminar vasoactive intestinal polypeptide (VIP) to layer 5 somatostatin (L5 SST) interneuron (IN) connections display different frequency-dependent short-term synaptic plasticity**

*A–C*, grand averages of IPSCs recorded in L5 SST IN using trains of five action potentials at 1 (*A*), 8 (*B*) and 40 Hz (*C*) presynaptic stimulation frequency (1 Hz: L5 PV to L5 SST *n* = 9, L5 VIP to L5 SST *n* = 11, L2/3 VIP to L5 SST *n* = 11; 8 Hz: L5 PV to L5 SST *n* = 8, L5 VIP to L5 SST *n* = 12, L2/3 VIP to L5 SST *n* = 12; 40 Hz: L5 PV to L5 SST *n* = 8, L5 VIP to L5 SST *n* = 12 L2/3 VIP to L5 SST *n* = 13). Averages of individual connections are shown in grey and the grand averages in blue (L5 PV to L5 SST), dark green (L5 VIP to L5 SST) or light green (L2/3 VIP to L5 SST).

were facilitating, which was already observable in the 2nd response and steadily increased to the 5th response (L5 VIP to L5 SST: normalized amplitude increase in the 2nd response: $28.63 \pm 35.72\%$, normalized amplitude increase in the 5th response: $82.35 \pm 73.72\%$, $n = 13$; L2/3 VIP to L5 SST: normalized amplitude increase in the 2nd response: $36.00 \pm 31.97\%$, normalized amplitude increase in the 5th response: $97.71 \pm 83.45\%$; $n = 13$).

Taken together PV and VIP IN effectively target L5 SST IN but display different unitary synaptic properties and short-term plasticities. These differences suggest distinct temporal and spatial modes of inhibition via these two connectivity motifs. Through both intralaminar and translaminar motifs L5 SST IN can be inhibited by a variety of local and long-range inputs to VIP IN, consistent with the notion of VIP IN as a major effector

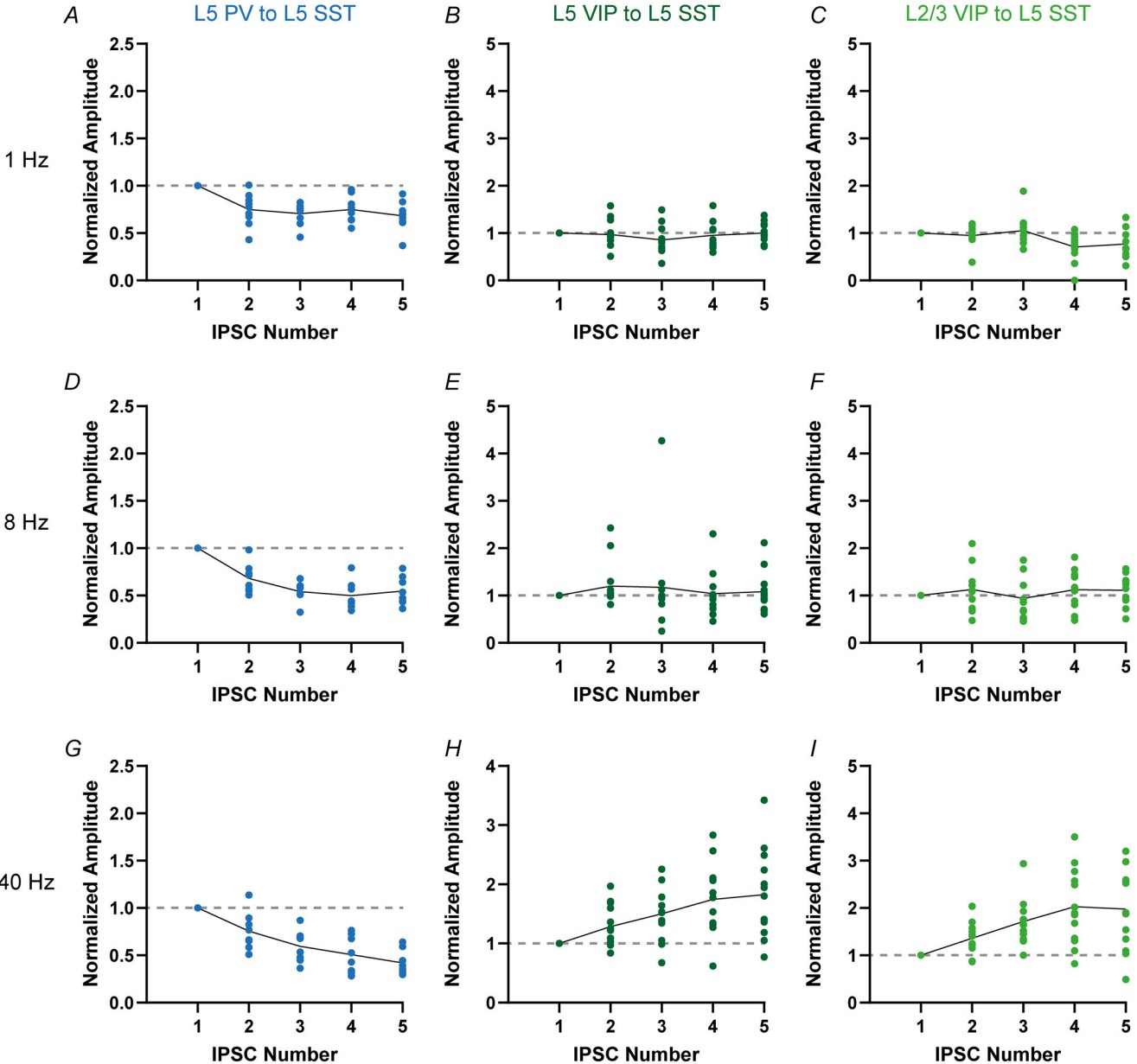

**Figure 8. Quantitative analysis of short-term synaptic plasticity at 1, 8 and 40 Hz presynaptic stimulation**
The amplitude ratio (amplitude of *n*th response divided by the amplitude of the first response) is shown. The layer 5 parvalbumin (L5 PV) to L5 somatostatin (L5 SST) motif (blue) displayed short-term depression at all three tested frequencies (1 Hz (*A*): $n = 9$; 8 Hz (*D*): $n = 8$; 40 Hz (*G*): $n = 8$). In contrast both vasoactive intestinal polypeptide (VIP) to L5 SST motifs displayed short-term facilitation but only at high-frequency firing (L5 VIP to L5 SST: dark green, 1 Hz (*B*): $n = 11$; 8 Hz I: $n = 12$; 40 Hz (*H*): $n = 12$; L2/3 VIP to L5 SST: light green, 1 Hz (*C*): $n = 11$, 8 Hz (*F*): $n = 12$, 40 Hz (*I*): $n = 13$). The continuous lines connect the means of each IPSC number.

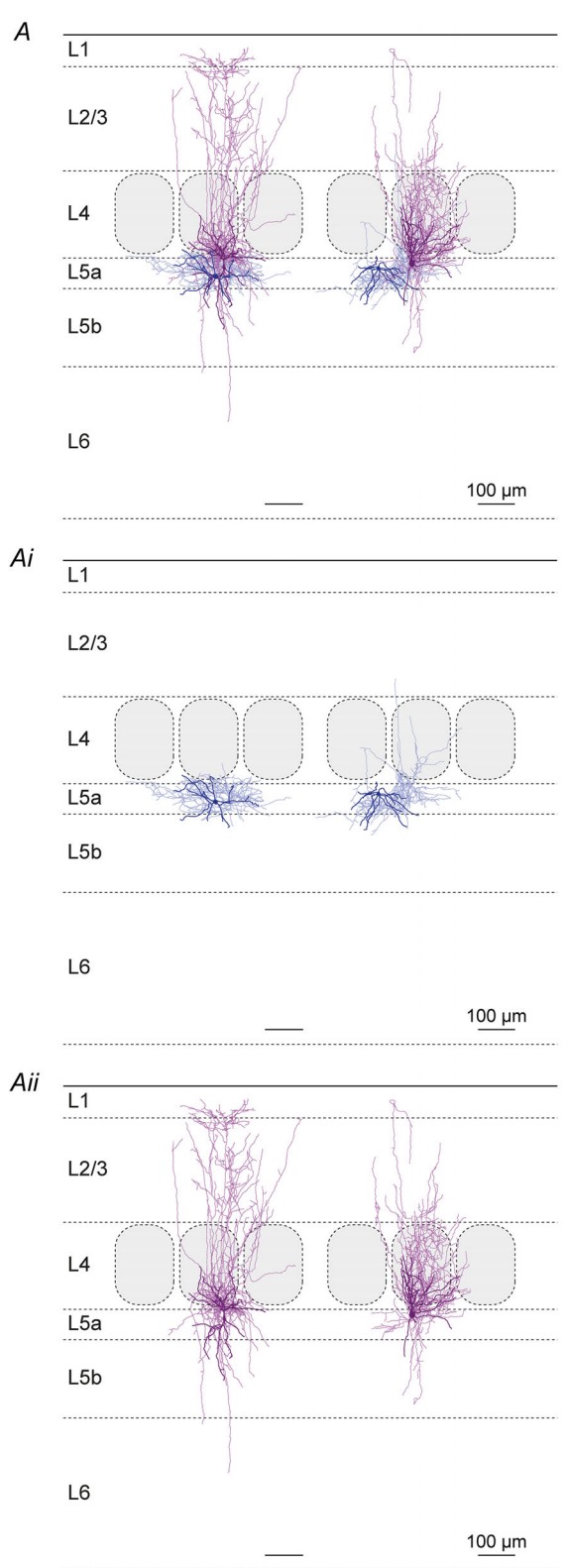

in light blue. To better display the details of individual cells PV and SST IN are shown separately in Ai and Aii. The left SST IN was classified as a Martinotti cell (MC) due to its axon branching out into L1. The axon of the right SST IN also reaches out into L1 but then descends back into L2/3 before any substantial innervation of L1 is established. Additionally the axon strongly innervates the home barrel. Therefore this cell was classified as nMC. Scale bar = 100 μm.

of disinhibition. However PV IN seem to target SST IN in a local manner, which establishes them as mediators of an intralaminar disinhibitory mechanism.

## Discussion

In this study we used triple-transgenic mouse lines to discover novel disinhibitory circuits in mouse wS1. The PV to SST and VIP to SST IN circuits have been described in multiple cortical areas (Fu et al., 2014; Jiang et al., 2015; Karnani et al., 2016; Lee et al., 2013; Pfeffer et al., 2013; Pi et al., 2013; Walker et al., 2016). However most studies are focused on L2/3, and little is known about L5, the main output layer is based on the canonical microcircuit (Adesnik & Naka, 2018) and the layer in which SST IN are predominantly located (Almasi et al., 2019). In addition studies that analysed translaminar circuits delivered varying results on the existence of such motifs (Campagnola et al., 2022; Jiang et al., 2015; Pfeffer et al., 2013). Here we were able to show that both L5 PV and VIP IN reliably target L5 SST IN, whereas only the translaminar L2/3 VIP to L5 SST motif displayed a substantial connectivity rate in contrast to a basically non-existent L2/3 PV to L5 SST IN connection. PV to SST and VIP to SST IN connections displayed IN type-specific differences in unitary synaptic properties and short-term plasticity. By combining optogenetics, paired recordings and morphological reconstructions, we thus were able to further extend our knowledge of the disinhibitory circuitry of mouse wS1. An important task is now to establish the *in vivo* functionality of intra- and translaminar circuits (Onodera & Kato, 2022; Senzai et al., 2019; Veit et al., 2023; Zhao et al., 2016), as features of whisker-dependent disinhibitory circuitry were discovered in recent two-photon-guided whole-cell recordings of L2/3 SST cells (Guy et al., 2023).

### Technical considerations

We used a caesium-based intracellular solution for postsynaptic SST IN. This approach comes with drawbacks, as we are not able to fully characterize the electrophysiological properties of these postsynaptic SST IN. According to the literature action potential firing properties might be another factor to distinguish between

**Figure 9. Gallery of morphologically reconstructed intralaminar layer 5 parvalbumin (L5 PV) to L5 somatostatin (L5 SST) interneuron (IN) pairs**
*A*, SST IN soma and dendrites are shown in dark purple and axon in light purple. PV IN soma and dendrites are shown in dark blue axon

different types of L5 SST IN (Ma et al., 2006; Nigro et al., 2018). A caesium-based intracellular solution also does not allow to further analyse the elementary and short-term plasticity properties of the reverse connections (SST to PV/VIP) that we observed in all four PV/VIP to SST connections. However we decided for a caesium-based solution, as we wanted to report the most accurate connection probability. In current literature connection probabilities vary depending on used intra-cellular solution (Campagnola et al., 2022; Walker et al., 2016). Comparing the connection probability of PV to SST IN in L2/3 of mouse V1 Campagnola et al. (2022) reported a connection probability of 9.1% (8/88) using a potassium gluconate-based internal solution, whereas Walker et al. (2016) reported a connection probability of 35.3% (in wS1) using a caesium-based solution and a holding potential of 0 mV. Together with our optogenetic experiments showing a very high abundance of these

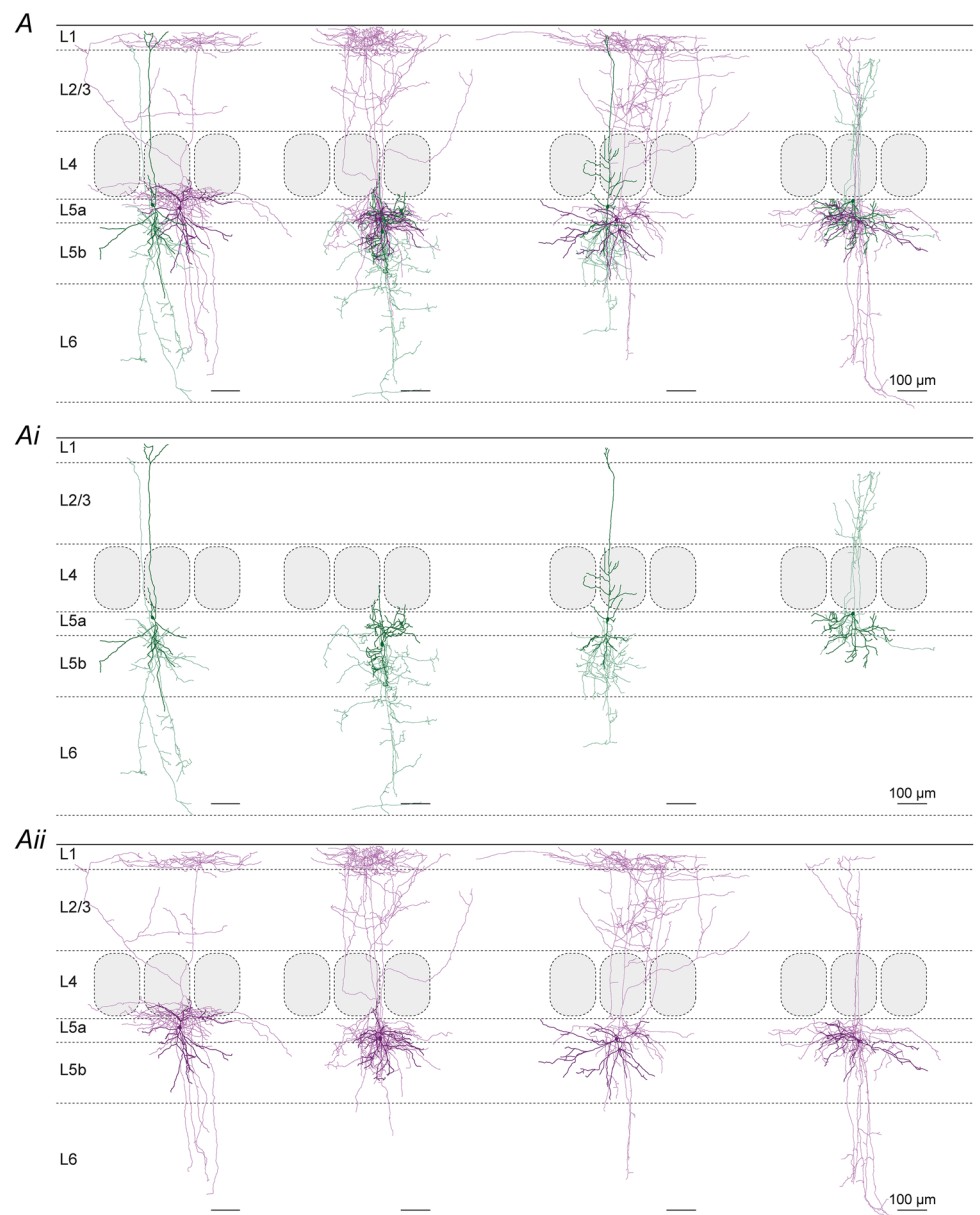

**Figure 10. Gallery of morphologically reconstructed intralaminar layer 5 vasoactive intestinal poly-peptide (L5 VIP) to L5 somatostatin (L5 SST) interneuron (IN) pairs**
*A*, SST IN soma and dendrites are shown in dark purple and axon in light purple. VIP IN soma and dendrites are shown in dark green and the axon in light green. To better display the details of individual cells VIP and SST IN are separately shown in Ai and Aii. The left three SST IN are clearly Martinotti cells (MC), whereas the right one is also considered as MC; however the pattern is disturbed by truncated L1-branching axon collaterals. Scale bar = 100 μm.

connections at the population level this points to the potential of strong undersampling of specific connections in an unbiased, large-scale high-throughput approach.

We used the GIN mouse line to genetically label SST IN. We are aware that the GIN mouse only sparsely labels SST IN in L5 but does that with high specificity (Oliva et al., 2000; Zhou et al., 2020). Most L5 GIN cells are located in L5a, which is why our dataset has a bias towards L5a.

Nevertheless we find different morphologically defined cell types in our dataset (nMC and MC including *t*-shaped and fanning-out MC) that are representative for the L5 SST IN population (Gouwens et al., 2019; Nigro et al., 2018). However we heavily rely on the successful morphological recovery of patched cells to assign our cells to the aforementioned three distinct cell types. Because we were only able to assign a cell type identity to ≈65% of

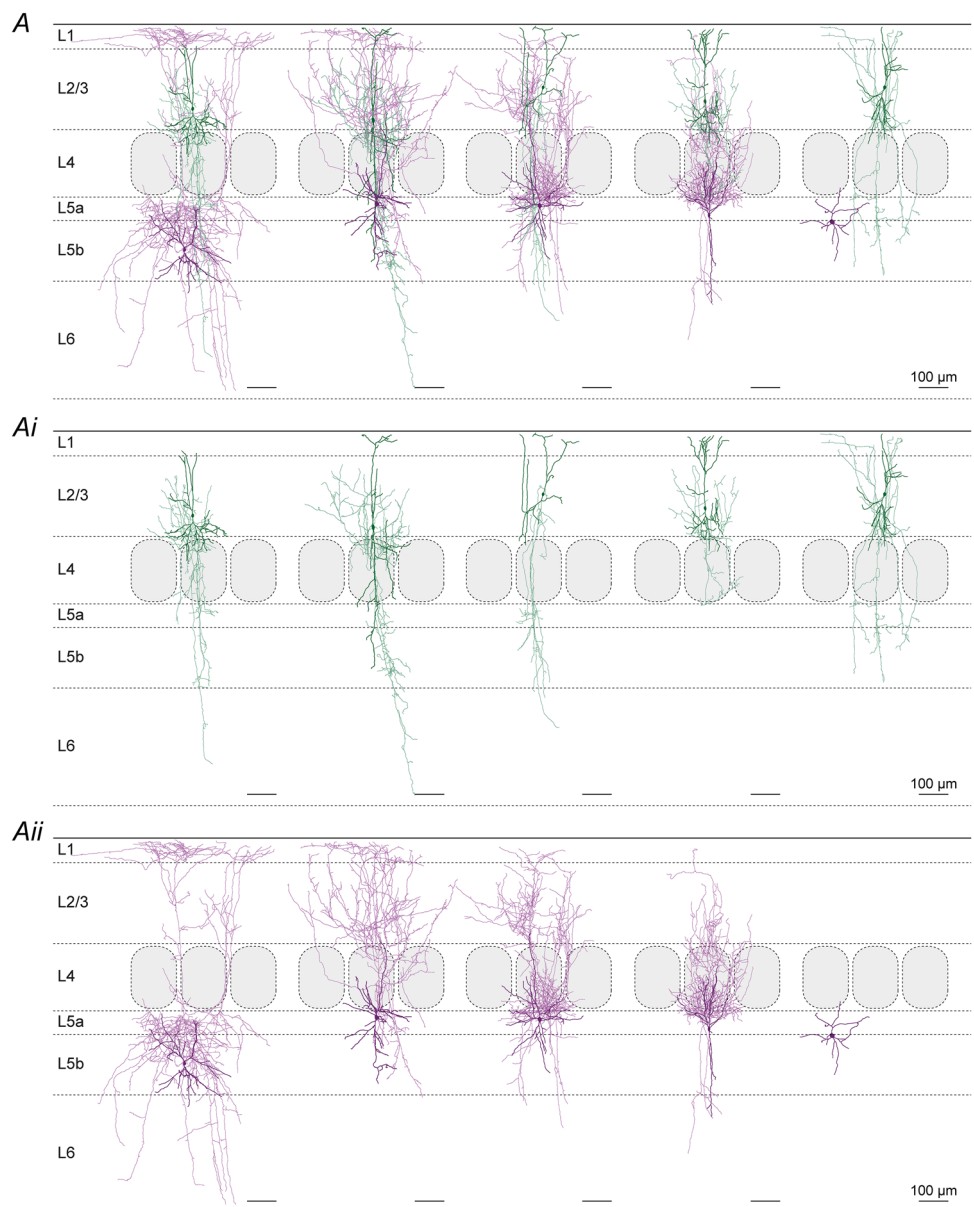

**Figure 11. Gallery of morphologically reconstructed translaminar layer 2/3 vasoactive intestinal polypeptide (L2/3 VIP) to layer 5 somatostatin (L5 SST) interneuron (IN) pairs**
*A*, Somatostatin interneurons (SST IN) soma and dendrites are shown in dark purple and axon in light purple. vasoactive intestinal polypeptide (VIP) IN soma and dendrites are shown in dark green and the axon in light green. To better display the details of individual cells VIP and SST IN are shown separately in Ai and Aii. The axon of the right SST IN has been cut shortly after the axon initial segment. Therefore only the somatodendritic configuration is shown for this cell. The three left SST IN are classified as Martinotti cells (MC) and the fourth from the left as non-MC (nMC). Scale bar = 100 μm.

our L5 SST IN we decided against splitting up our dataset into multiple subgroups of SST IN for maintaining a high power in our statistical analysis. This means that we might have missed putative SST subtype-specific differences. In a recent study Wu and colleagues demonstrated multiple novel approaches to target genetically defined subtypes of SST IN with, by and large, distinct axonal projection patterns (Wu et al., 2023). In addition they showed that SST subtypes target different populations of PN. It would be interesting to see whether these SST subtypes are also targeted differently by PV and VIP IN. The use of genetically defined SST 'intersectional' subtypes may also alleviate the problem that unrecovered cells can hardly be assigned to a distinct (morphologically defined) cell type.

Paired recordings are the gold standard in the field to probe monosynaptic connections. To the best of our knowledge inducing firing in a presynaptic inhibitory IN has never been shown to result in a disynaptic response in a target cell. Thus the relatively long latencies found in some L2/3 VIP to L5 SST IN pairs (Fig. 6*F*) are very likely attributable to a more distant synapse location (see also below).

## Cell type-specific differences in inhibitory PV and VIP inputs to SST IN

The cortical disinhibitory circuitry (especially the VIP to SST motif) has been extensively studied for over a decade (Campagnola et al., 2022; Jiang et al., 2015; Pfeffer et al., 2013; Pi et al., 2013; Williams & Holtmaat, 2019). However most of these studies focused on disinhibitory circuits of supragranular layers. Little is known about disinhibitory circuits of L5 in which SST IN are predominantly located (Almasi et al., 2019; Zhou et al., 2017). Large-scale studies that also analysed infragranular layers are either contradictory or were unable to successfully record VIP to L5 SST connections (Campagnola et al., 2022; Jiang et al., 2015). In this study we were able to reveal two novel intra-laminar (L5 PV and L5 VIP to L5 SST) and one novel translaminar (L2/3 VIP to L5 SST) disinhibitory circuit motif in mouse wS1. All three elementary connections demonstrated IN type-specific differences in unitary synaptic properties and short-term synaptic plasticity. The PV to SST connection was larger in amplitude, shorter in latency and had a larger synaptic success rate in comparison to VIP connections. The cell type-specific differences that we observed might be explained by different subcellular synaptic targeting (Feldmeyer et al., 2018; Staiger & Petersen, 2021). Although PV IN are thought to control the output of their target cells via perisomatic inhibition (Freund & Katona, 2007), VIP IN synapses are considered to mostly target intermediate calibre dendritic branches ($\approx$80% of all VIP synapses) (Zhou et al., 2017). Because we were recording at the soma, synaptic inputs at distal dendrites are attenuated and delayed (Spruston et al., 1993), which might explain the prolonged latency and reduced amplitude and synaptic success rate that we observed in both L5 VIP to L5 SST and L2/3 VIP to L5 SST pairs. Other underlying mechanisms might be different in GABA$_A$ receptor phosphorylation (Hinkle & Macdonald, 2003; Kapur & Macdonald, 1996; Poisbeau et al., 1999), GABA lifetime in the synaptic cleft (Nusser et al., 2001) or GABA$_A$ receptor subunit composition (Bacci et al., 2003; Huang et al., 2023).

In addition we report IN type-specific differences in short-term synaptic plasticity. PV inputs showed synaptic depression at all tested frequencies, whereas VIP inputs were facilitating but only at high-frequency stimulation. In general PV IN inputs are depressing, independent from their postsynaptic target cell (Ma et al., 2012). This was also confirmed in a recent study from visual cortex (Campagnola et al., 2022). Nonetheless synaptic depression of PV to SST inputs at 50 Hz stimulation as shown by Campagnola et al. (2022) was smaller than in our study at 40 Hz stimulation ($-33$%) for reasons to be determined. Campagnola and colleagues also analysed VIP to SST connections but only found a minor effect ($+5$% amplitude increase) on amplitude size. However in the latter study PV, SST and VIP IN, respectively, were pooled for analysis and not apportioned based on their layer location. Our cell type-specific differences are similar to one of our previous studies in which we analysed intra-laminar PV and VIP to SST IN connections in L2/3 of wS1 (Walker et al., 2016), suggesting that these properties are conserved across layers.

Current hypotheses posit that disinhibition improves information processing in pyramidal cell networks (Kullander & Topolnik, 2021). Given that VIP to SST IN connections show strong facilitation at high-input frequencies, whereas PV to SST IN connections substantially depress at all frequencies, a high-input regime on SST IN by VIP IN could ensure disinhibition to be ramped up over longer time scales than those mediated by PV IN. Thus disinhibition could sustain longer when VIP IN, rather than PV IN, are active as disinhibitors (Fuhrmann et al., 2002; Tremblay et al., 2016).

## Translaminar L2/3 PV to L5 SST connections are extremely rare

Albeit not as prominent as in L2/3 VIP IN (Prönneke et al., 2015) multiple studies described descending axon collaterals in L2/3 PV cells (Gouwens et al., 2019; Jiang et al., 2015; Kubota, 2014). Therefore we wanted to test whether L2/3 PV IN also target L5 SST IN. We only found a single connected L2/3 PV to L5 SST IN pair when unbiasedly choosing presynaptic PV IN. Nevertheless with 36.6% the reverse connectivity of L5 SST to L2/3 PV

IN was the highest observed in this study, further underlining the emerging prominence of this circuit motif that has also been observed *in vivo* (Cottam et al., 2013). Therefore our findings cannot be explained by slicing artefacts or other technical errors. The absence of a potent L2/3 PV to L5 SST connection is in line with other studies. Jiang et al. (2015), for example, analysed the connectivity of more than 1200 IN in L1, L2/3 and L5 in mouse V1. They also found an extremely low connectivity rate of L2/3 BC to L5 MC (0.8% (1/123)). As target cells in L5 they were able to identify other BC cells (13.6%), PN (5.9%) and horizontally elongated cells (4.2%) (Jiang et al., 2015). Jiang and colleagues used morphology, and not marker expression, to classify their cells, thereby separating different SST IN types into different morphological classes, for example, as MC, BC or bitufted cells (BTC), which makes it difficult to precisely compare their findings to ours, focusing on SST IN identified through transgenic marker expression.

We suggest that translaminar L2/3 PV to L5 SST pairs might be developmental remains (Kalisman et al., 2005). Micheva and colleagues reported a massive PV synapse pruning in neocortex with a threefold decrease in synapses in mice from P14 to 1 month of age (Micheva et al., 2021). They analysed PV to PN pairs but did not report an overall connectivity. We performed paired patch-clamp recordings in young adult mice (age for tested L2/3 PV to L5 SST connections: median P36.5, minimum P29, maximum P57). It would be interesting to test whether L2/3 PV to L5 SST connections are more abundant in developing mice (P7–14), which might be a hint to a developmental reduction in translaminar PV to SST IN connections via synapse pruning.

### Role of PV and VIP IN as disinhibitors in information processing

PV IN are the largest population of neocortical GABAergic IN (≈40%) and can be subdivided into chandelier (or axo-axonic) cells and BC (Inan & Anderson, 2014; Staiger & Petersen, 2021; Tremblay et al., 2016). Due to the target specificity of chandelier cells by forming synapses on the axon initial segments of PN (Somogyi, 1977) BC are the relevant PV subtype in the present study. BC are suggested to sample activity from local excitatory cells and provide feedforward and feedback inhibition on the local microcircuit (Meeuwissen et al., 2023; Packer & Yuste, 2011; Staiger & Petersen, 2021). Our data suggest that the disinhibitory effect of PV IN is probably focused or even restricted to intralaminar target cells. Furthermore the PV to SST connection seems to be strong, fast and reliable but only at the beginning of activity due to the depressing synaptic plasticity and the high interconnectivity of PV IN. This might allow a different temporal window of opportunity for inhibition of target cells by PV *versus* VIP IN (Walker et al., 2016).

Although only ≈12% of GABAergic IN express VIP (Pfeffer et al., 2013; Tremblay et al., 2016), recent single-cell RNA-seq studies suggest up to 16 different potential VIP transcriptomic types (Tasic et al., 2018). Intersectional breeding suggests to differentiate between VIP IN coexpressing calretinin (CR) and cholecystokinin (CCK), as both intersectional lines differ in morphology, cortical layer location, electrophysiological properties and target specificity (Guet-McCreight et al., 2020; He et al., 2016; Paul et al., 2017). These authors propose that multipolar VIP/CCK IN correspond to the subpopulation of VIP and CCK-expressing small BC, whereas most VIP/CR IN were bipolar and are considered to be interneuron targeting. Overall VIP IN are known for their role as disinhibitors in cortical circuitry, especially the VIP to SST connection is the most prominent disinhibitory circuit motif (Fu et al., 2014; Lee et al., 2013; Pfeffer et al., 2013). We were able to show that L5 VIP IN in wS1 generate feedforward inhibition onto intralaminar SST IN, as it was recently reported that paralemniscal input excites L5 VIP IN (Audette et al., 2018; Sermet et al., 2019). In addition we described a novel translaminar L2/3 VIP to L5 SST circuit motif. Here the possibilities of signal integration are much more complex. L2/3 VIP IN have been shown to receive direct input from primary motor cortex (M1) (Lee et al., 2013; Naskar et al., 2021), cholinergic inputs from basal forebrain (Fu et al., 2014; Gasselin et al., 2021) and are directly involved in sensory information processing (Guy et al., 2023; Ramamurthy et al., 2023). The L2/3 VIP to L5 SST projection that we describe here would allow multiple local and distant input sites to reduce the inhibitory drive onto L5 PN via disinhibition of L5 SST IN through L2/3 VIP IN activation; therefore it would enable L5 PN to send processed sensory information to various other locations within the brain.

### Role of reciprocal connections

In all four analysed connectivity motifs we found a high probability (≈30%–50%) of reciprocal (or reverse) connections. These reciprocal connections offer new possibilities of how sensory stimuli are processed in the cortex and also raise the question which IN are activated first to predict the direction of signal flow and the mode of sensory information processing (Kepecs & Fishell, 2014). A recent *in vivo* study showed that VIP IN were more rapidly excited during free whisking-in-air but responded to active touch after a delay (Kiritani et al., 2023). This would open a window of opportunity for SST IN to 'inhibit their own inhibitors'. Another *in vivo* example is the optotagging work of Muñoz et al. in which they analysed

the activity of SST subtypes. They found a population of SST IN in infragranular layers that were depolarized upon whisking (Muñoz et al., 2017). Morphologically these IN were either defined as fanning-out MC or nMC, two subtypes that we also observed in our dataset. Because SST IN only receive weak inputs from motor cortex (Lee et al., 2013) and thalamus (Cruikshank et al., 2010), Muñoz and colleagues suggest that these inputs cannot be the source of SST activation upon whisking. They argued that the excitatory drive onto SST IN is coming from acetylcholine release from basal forebrain afferents, which are also strongly correlated with locomotion and whisking (Eggermann et al., 2014; Nelson & Mooney, 2016). Taken together timed basal forebrain activity might be a possible mechanism of how reciprocally connected SST IN are able to actively inhibit their own inhibitors, allowing differential integration of basic circuit motifs in cortex based on attention status.

In summary we reported two novel intralaminar and one novel translaminar disinhibitory circuit in mouse wS1. PV to SST and VIP to SST connections revealed IN type-specific differences in unitary synaptic properties and short-term synaptic plasticity. The VIP IN targeting by multiple long-range afferents and local sensory inputs (Naskar et al., 2021) allows VIP IN to process information from multiple sources and to integrate this information into somatosensory processing, potentially by co-ordinating ensembles through gamma rhythms (Veit et al., 2023).

## Materials and methods

### Mouse lines and ethical approval

All experiments were conducted in accordance with the German guidelines of animal care. The experimental protocol was approved by the Niedersächsische Landesamt für Verbraucherschutz und Lebensmittelsicherheit (LAVES; 21/3656). All animals were obtained from The Jackson Laboratory (Bar Harbor, USA). Mice were housed under standard cage conditions with access to food and water *ad libitum* and on a 12 h light/dark cycle. PV-Cre (B6;129P2-Pvalbtm1(cre)Arbr/J, strain #008069) or VIP-Cre (VIPtm1(cre)Zjh/J, strain #010908) mice were cross-bred with Ai9 mice (B6.Cg-Gt(ROSA)26Sortm9(CAG-tdTomato)Hze/J, strain #007909) to generate PV-Cre//tdTomato and VIP-Cre//tdTomato mice. Those were further cross-bred with GIN mice (FVB-Tg(GadGFP)45704Swn/J, #003718) to generate triple-transgenic PV-Cre//tdTomato//GIN and VIP-Cre//tdTomato//GIN mice. In these mouse lines PV- and VIP-expressing IN can be identified by their tdTomato and GIN cells by their GFP fluorescence. GIN stands for GFP-expressing IN, and EGFP was found to be expressed in a subpopulation of SST-expressing IN (Oliva et al., 2000). Because 100% of EGFP cells also express SST (Zhou et al., 2020) we are referring to GIN cells as SST IN.

### AAV injection

To test whether L5 SST IN receive inputs from PV and VIP IN we stereotactically injected pAAV6-EF1a-double floxed-hChR2(H134R)-mCherry-WPRE-HGHpA into mouse barrel cortex. This AAV was custom manufactured by the 'Viral Vectors Platform' of the DFG research unit and cluster of excellence CNMPB Goettingen and causes Cre-dependent expression of ChR2 and mCherry. PV-Cre//GIN or VIP-Cre//GIN mice (postnatal days, 18–34, both sexes) were anaesthetized using 3% isoflurane (1%–2.5% isoflurane during surgery) and placed in a stereotactic frame (Kopf Instruments, Tujunga, CA, USA). For preoperative analgesia buprenorphine (0.65 µg/g, i.p.) was administered. The skull was exposed, and a small craniotomy was performed (ca. 1 mm in diameter) with a dental drill (Osada Success 40, Osada, Tokyo, Japan). AAVs were injected into the barrel cortex (co-ordinates: anterior–posterior range: Bregma minus 1–2 mm; medial-lateral range: 2.5–3.5 mm) using glass micropipettes cut to 20 µm diameter connected to a Toohey Spritzer Pressure System IIe (Toohey Company, Fairfield, NJ, USA). AAVs were injected at two to five different locations based on blood vessel patterns. ChR2-encoding AAV was injected at three different depths (250, 500 and 750 µm) below pia surface. Ca. 100–150 nl of virus per position and depth (diluted in sterile PBS) were injected using pressure pulses (3 psi, 250 ms pulse duration). The micropipette was retracted 5–10 min after virus application. Mice were sutured and received carprofen (5 µg per g body weight, subcutaneous) directly after surgery and after 24 and 48 h.

### Brain slice preparation

Mice of both sexes (postnatal days 23–57) were deeply anaesthetized with isoflurane and decapitated. Brains were quickly removed and transferred into the ice-cold cutting solution (in mM: 87 NaCl, 1.25 $NaH_2PO_4$, 2.5 KCl, 10 glucose, 75 sucrose, 0.5 $CaCl_2$, 7 $MgCl_2$ and 26 $NaHCO_3$; pH 7.4), which was constantly oxygenated (carbogen: 95% $O_2$/5% $CO_2$). Hemispheres were separated, and 300 µm thalamocortical slices were cut according to Porter et al. (2001). Slices containing the barrel field were transferred to oxygenated artificial cerebral and spinal fluid (ACSF) (in mM: 125 NaCl, 1.25 $NaH_2PO_4$, 2.5 KCl, 25 glucose, 2 $CaCl_2$, 1 $MgCl_2$, 26 $NaHCO_3$) and incubated at 32°C for 30–45 min. Afterwards slices were kept at room temperature until further use.

### Electrophysiology and data acquisition

Slices were transferred to a submerged recording chamber with a constant flow of ACSF (flow rate ca. 2 ml min$^{-1}$ at 32°C) in an upright microscope (Axio Examiner, Zeiss, Germany). Recording pipettes (5–8 MΩ) were pulled from borosilicate glass (Science Products, Hofheim, Germany) using a P-1000 Micropipette Puller (Sutter Instruments, Novato, USA). For GIN cells in paired recordings a caesium-based intracellular solution (containing in mM: 135 CsMeSO$_4$, 5 CsCl, 0.5 EGTA, 10 Hepes, 4 Mg-ATP, 0.3 Na-ATP, 10 Na-phosphocreatine phosphate, pH: 7.4) was used. For presynaptic PV and VIP IN or characterization of individual GIN cells a potassium gluconate-based solution (containing in mM: 135 K-gluconate, 5 KCl, 0.5 EGTA, 10 HEPES, 4 Mg-ATP, 0.3 Na-ATP, Na-phosphocreatine-phosphate, pH: 7.4) was used. Before experiments 0.3%–0.5% biocytin was added to internal solutions to allow morphological reconstruction of patched cells. Data were acquired using SEC-05X amplifiers with a SEC-05X-BF low-noise head stage (npi Electronics, Tamm, Germany) in discontinuous mode with a switching frequency of 50 kHz. Signals were filtered at 3 kHz and digitized at 10–25 kHz using a CED Power 1401 interface (CED Limited, Cambridge, England). Whole-cell recordings were performed in current clamp and voltage clamp mode. For cell characterization hyperpolarizing and depolarizing current pulses were used. Data were collected using Signal 5 (CED Limited, Cambridge, England).

### Optogenetic stimulation and recording

Two to 3 weeks after virus injection animals were killed, and acute brain slices prepared as described above. For photostimulation of ChR2 a 473 nm laser was used (DL-473, Rapp Opto Electronics, Wedel, Germany). SST IN in L5 were patched with caesium-based intracellular solution and held at 0 mV. To test whether they receive inputs from PV and VIP IN in general we stimulated a ≈100 μm diameter spot with the SST soma in the centre for 1 ms. We recorded three different intensities: sub-threshold, threshold and 10 × threshold at least thrice with a 5 s interstimulus-interval.

### Paired recordings

For intralaminar paired recordings we recorded from IN in close proximity (ca. 20–200 μm). In case of trans-laminar paired recordings we mostly recorded from IN that were located in the same cortical column. Presynaptic PV or VIP IN were recorded at their resting membrane potential ($V_{rest}$). Postsynaptic GIN cells were held at 0 mV in voltage clamp to increase the driving force for inhibitory chloride currents. For induction of action potentials in the presynaptic cells a short square pulse (5 ms, 10 s delay between sweeps, 10–30 sweeps) was used. In some cases we used a short test pulse (200 ms, −5 mV) in postsynaptic SST IN to analyse the access resistance ($R_a$). In 31 tested connected SST IN access resistance was 14.04 ± 0.76 MΩ (mean ± s.e.m.). To investigate the short-term synaptic plasticity a train of five action potentials at 1, 8 and 40 Hz stimulation was used. If cells were still stable after the recordings, we also checked for the reverse connections (SST to PV, SST to VIP) to test whether these cells are reciprocally connected.

For analysis custom written scripts for Signal 5 were used. Only traces with responses in postsynaptic cells were aligned with regard to the presynaptic action potential peak and averaged. The following parameters have been analysed: amplitude (difference from base-line to peak), latency (time from presynaptic spike peak to IPSC onset (>3× standard deviation)), 10%–90% rise time (time between 10% and 90% IPSC rise) and normalized slope of the rising IPSC phase (average slope, determined by means of a least-square fit, divided by the maximum amplitude). For short-term synaptic plasticity experiments we only analysed IPSC amplitudes and calculated the amplitude ratios. Therefore amplitudes were divided by the amplitude of the first response. IPSC overlaps were only observed at 40 Hz stimulation. In that case the decay phases of IPSCs were exponentially fitted and extrapolated to baseline level. Amplitudes were defined as distance from IPSC peak to a fitted curve of previous IPSC.

### Immunohistochemistry

After experiments slices were fixed in 4% paraformaldehyde (PFA) + 15% picric acid in phosphate-buffer (PB; 0.1 M, pH 7.4) at 4°C overnight. Slices were washed at least 5 times with PB buffer for 15 min to fully remove PFA and picric acid. Slices were then washed in TRIS buffer (TB; 2 × 15 min; 0.05 M, pH 7.6), TRIS-buffered saline (TBS; 2 × 15 min), TBS + 0.5% Triton-X 100 (TBST; 2 × 15 min). Unspecific binding sites were blocked using 0.25% bovine serum albumin + 10% normal donkey serum for 90 min. Slices were incubated with primary antibodies (goat $\alpha$-GFP, 1:2000 (Abcam, Cambridge, UK); rabbit $\alpha$-RFP, 1:500 (Rockland, Limerick, PA, USA)) for 48–72 h at 4°C. Afterwards they were washed with TBST (4×15 min) and incubated with secondary antibodies (donkey $\alpha$-goat AF488 and donkey $\alpha$-rabbit AF546 (both 1:500, Molecular Probes, USA)) and AF633-conjugated streptavidin (1:300) in TBST for 4 h. Nuclei were visualized using DAPI (1:5000) in TBS buffer for 5 min. Finally slices were washed with TBS (1 × 15 min) and TB buffer (2 × 15 min) before they were mounted in AquaPolyMount (Polysciences, Warrington,

PA, USA) and covered with a cover-slip (24 × 5 mm, 0.08–0.12 mm thickness (Menzel)).

### Reconstruction of neurons

Overview images (10× magnification) of all slices were taken using an inverted microscope (Axio Observer, Zeiss, Oberkochen, Germany). If both IN recovered reasonably without too much background stained we took image stacks of the whole cortical volume in which we could observe neurites using a confocal microscope (Zeiss LSM 880). We used different objectives (40× water or 63× oil) and either classical confocal microscopy or Zeiss Airyscan fast mode (Huff, 2015). Images were stitched using Zen Black software (Zeiss) or Grid/Collection stitching plugins in Fiji (Preibisch et al., 2009). Reconstruction of neurons was performed using Neurolucida software (MBF Bioscience, Colchester, USA).

### Statistics

For statistical comparison GraphPad Prism 9 or 10 was used (GraphPad Software, Boston, MA, US). All groups were tested for normality using Shapiro–Wilk test. For three groups and no normal distribution the Kruskal–Wallis test with Dunn's multiple comparison was used. If the data were normally distributed an ordinary ANOVA with Tukey's multiple comparison test was used. For comparison of two groups with normal distribution an unpaired t test was used, otherwise the Mann–Whitney test was used. Data are described as mean ± SD (if not stated otherwise) and visualized as 10%–90% box and whisker plots. $P$-values $<0.05$ were interpreted as significantly different.

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

## Additional information

### Data availability statement

The data that support the findings of this study are available from the corresponding author on request.

### Competing interests

The authors declare that they have no competing interests.

### Author contributions

F.P., F.W. and M.W. carried out experiments. F.P., F.W., M.M., M.W. and J.F.S. were involved in the study design. F.P. and J.F.S. wrote the manuscript. M.W. and J.F.S. supervised all experiments. F.P., F.W., M.W. and M.M carried out data analysis.

### Funding

This study was supported by grants from the German Research Foundation (DFG, STA 431/14-1; 21-1; WI 5636/2-1).

### Acknowledgements

We thank Patricia Sprysch, Sandra Heinzl and Pavel Truschow for their excellent technical assistance and Sophia Heidenreich, Sabrina Hübner, Ima Mansori, Leander Matthes, Paul Molis and Nicolas Zdun for morphological reconstructions.

## Keywords

cortical circuits, disinhibition, parvalbumin, short-term plasticity, somatosensory cortex, somatostatin, vasoactive intestinal polypeptide

## Supporting information

Additional supporting information can be found online in the Supporting Information section at the end of the HTML view of the article. Supporting information files available:

**Peer Review History**

