## [Peer Review History · The Journal of Physiology]

Inhibitory circuit motifs of cortical somatosensory layer 5 SST interneurons are uniform within layers but specific across layers

Felix Preuss, Florian Walker, Martin Möck, Mirko Witte, and Jochen Staiger

DOI: 10.1113/JP288309

Corresponding author(s): Jochen Staiger (jochen.staiger@med.uni-goettingen.de)

Review Timeline:

Submission Date:	13-Dec-2024
Editorial Decision:	17-Mar-2025
Revision Received:	02-Jul-2025
Editorial Decision:	26-Aug-2025
Revision Received:	27-Oct-2025
Accepted:	31-Oct-2025

Senior Editor: Katalin Toth

Reviewing Editor: Conny Kopp-Scheinflug

Transaction Report:

Dear Dr Staiger,

Re: JP-RP-2024-288309 "Uniform intralaminar but specific translaminar disinhibitory circuit motifs of L5 SST neurons in mouse barrel cortex" by Felix Preuss, Florian Walker, Martin Möck, Mirko Witte, and Jochen Staiger

Thank you for submitting your manuscript to The Journal of Physiology. It has been assessed by a Reviewing Editor and by 2 expert referees and we are pleased to tell you that it is potentially acceptable for publication following satisfactory major revision.

REVISION CHECKLIST:

Please upload two versions of your manuscript text: one with all relevant changes highlighted and one clean version with no

changes tracked. The manuscript file should include all tables and figure legends, but each figure/graph should be uploaded as separate, high-resolution files.

We look forward to receiving your revised submission.

Yours sincerely,

Katalin Toth
Senior Editor
The Journal of Physiology

REQUIRED ITEMS

- Include a Key Points list in the article itself, before the Abstract.

- Author photo and profile. First or joint first authors are asked to provide a short biography (no more than 100 words for one author or 150 words in total for joint first authors) and a portrait photograph. These should be uploaded and clearly labelled together in a Word document with the revised version of the manuscript. See Information for Authors for further details.

- Your manuscript must include a complete Additional Information section, including competing interests; funding; author contributions and acknowledgements.

- Please upload separate high-quality figure files via the submission form.

- Please ensure that any tables are editable and in Word format, and wherever possible, embedded in the article file itself.

- Your paper contains Supporting Information of a type that we no longer publish, including supplementary tables and figures. Any information essential to an understanding of the paper must be included as part of the main manuscript and figures. The only Supporting Information that we publish are video and audio, 3D structures, program codes and large data files. Your revised paper will be returned to you if it does not adhere to our Supporting Information Guidelines.

- Papers must comply with the Statistics Policy: https://jp.msubmit.net/cgi-bin/main.plex?form_type=display_requirements#statistics.

In summary:

- If $n \leq 30$, all data points must be plotted in the figure in a way that reveals their range and distribution. A bar graph with data points overlaid, a box and whisker plot or a violin plot (preferably with data points included) are acceptable formats.

- If $n > 30$, then the entire raw dataset must be made available either as supporting information, or hosted on a not-for-profit repository, e.g. FigShare, with access details provided in the manuscript.

- 'n' clearly defined (e.g. x cells from y slices in z animals) in the Methods. Authors should be mindful of pseudoreplication.

- All relevant 'n' values must be clearly stated in the main text, figures and tables.
- The most appropriate summary statistic (e.g. mean or median and standard deviation) must be used. Standard Error of the Mean (SEM) alone is not permitted.
- Exact p values must be stated. Authors must not use 'greater than' or 'less than'. Exact p values must be stated to three significant figures even when 'no statistical significance' is claimed.

- Please include an Abstract Figure file, as well as the Figure Legend text within the main article file. The Abstract Figure is a piece of artwork designed to give readers an immediate understanding of the research and should summarise the main conclusions. If possible, the image should be easily 'readable' from left to right or top to bottom. It should show the physiological relevance of the manuscript so readers can assess the importance and content of its findings. Abstract Figures should not merely recapitulate other figures in the manuscript. Please try to keep the diagram as simple as possible and without superfluous information that may distract from the main conclusion(s). Abstract Figures must be provided by authors no later than the revised manuscript stage and should be uploaded as a separate file during online submission labelled as File Type 'Abstract Figure'. Please also ensure that you include the figure legend in the main article file. All Abstract Figures should be created using BioRender. Authors should use The Journal's premium BioRender account to export high-resolution images. Details on how to use and access the premium account are included as part of this email.

EDITOR COMMENTS

Reviewing Editor:

Dear Dr. Staiger and co-authors,

your manuscript has been reviewed by two experts in the field. As you will see both rate the quality of the recordings and reconstructions as excellent. However, the reviews also point out that the manuscript needs a major re-write to streamline the flow of information. Reviewer 1 in particular gave some detailed examples of how to address this task. Please carefully revise your manuscript accordingly.

REFEREE COMMENTS

Referee #1:

This is a very focused study providing high-quality patch-clamp recording data regarding inhibitory control of L5 somatostatin (SST) neurons. The major finding is that SST neurons are inhibited by VIP neurons regardless of layer, but by PV cells only when they reside in layer 5. The reconstructions are excellent and will contribute to databases of cellular anatomy and connectivity if made freely available. Some analysis of short term plasticity is also provided, although the significance of this measurement is unclear. One major limitation is that the diversity of L5 SST and also that of VIP neurons is not explored, and this might have an important role in establishing specific motifs for VIP and SST inhibition. Although the authors focused on SST neurons that are labeled in the GIN line, this may exclude some subtypes of SST neurons found in L5. Finally, authors indicate that these circuits are novel - this is not strictly true, as these motifs have been well-defined in multiple other studies. However, the specific details of connectivity presented here are rigorous acquired and evaluated and will be useful to experts in this subfield.

In general, the data are strong and the results convincing. However, the manuscript is written in a complex and convoluted manner that makes extracting the major conclusions difficult and detracts from the main message. This is the case throughout, from the title to the abstract and throughout the body of the text. Some recommendations about wording and vocabulary are provided below. There are also multiple tangents that detract from the main message. As it stands, the work is not ready for publication.

Main critique

1. The title is convoluted and difficult to understand.
2. "Selectivity in translaminar disinhibitory targeting" is unnecessarily complicated. Referring to disinhibition is not really

necessary in the context of this paper, since the authors are not really looking at effects on network activity of pyramidal neurons. A better focus might be the generalized finding that VIP neurons inhibit L5 SST neurons, regardless of what layer they reside in. PV neurons can serve as a contrast to this, where their inhibition is local.

3. Throughout: "Besides" is not typically the right word. "In addition" would be better.

4. The introduction is weighed down by an unnecessary description of the vibrissae system in rodents and the 6 layers of the neocortex. This could be removed without changing the significance of the manuscript. Likewise, all the references about the "important role" that the 6-layered cortex might perform in "information processing." This is vague and also not the subject of the present study. The authors are encouraged to make reference to specific inhibitory motifs that have been proposed in other studies, and indicate how their data help revise those models.

5. Please do not refer to "excitation-inhibition balance." This is an outdated concept of limited utility in thinking about specific networks of connectivity for controlling the activity of inhibitory neurons. Similarly, the term "disinhibitory circuitry" is unnecessarily broad and also conceptually complex - why not just talk about inhibition onto SST? Disinhibition as a topic is too expansive, because there will also be inhibition onto PV neurons, and inhibition onto VIP, two topics that are not explored in the manuscript.

6. References are idiosyncratic - why is Nigro 2018 cited to refer to non-L1 projecting axons? This was described significantly earlier by Agmon and colleagues. Line 68 - references about inputs to L5 neurons are similarly oddly selected or not specific to the point being made (Yao 2023?) Line 72 - why is Gouwens et al 2019 or Wu ... Fishell not cited? Line 309 - why is Pfeffer ... Scanziani or Jiang ... Tolias not cited? Line 464 - Audette ... Barth also showed this motif. Line 466 - Naskar ... Lee also rigorously examined this.

7. The focus on L5a but not L5b SST neurons is unclear. Why was this done? Was the distribution of Martinotti and non-Martinotti different across L5a and 5b?

8. Data showing that L5 SST neurons have similar electrophysiological properties despite different axonal morphologies is useful. This point should be highlighted more. When reconstructions are available, the authors should show whether VIP and PV inhibition has any specificity to these two proposed subtypes. Was there any difference in rebound spikes, given a hyperpolarizing pulse, as has been suggested in some studies? Since there is some controversy about how many different subtypes of SST neurons there might be, it would be useful to the field to try to address this with the current data. Since morphological data were obtained for 65% of SST neurons targeted, the authors should at least assess the properties of different SST morphologies (fanning or T-shaped).

9. The number of observations (n's) is hard to find in many places. It would be easiest to put this in the figures, or if the data are not shown in figure then in the text, for each value presented.

10. The discussion is quite elaborate and it is clear that the authors are very familiar with a large body of work. However, much of the discussion is overly speculative and not fully justified by the data presented (see line 494-6 - this is highly speculative, and the cited studies are not focused on L5 SST neurons). See also Line 517-520 - again, highly speculative and not justified by the data presented in the paper.

Minor corrections:

1. Line 115 - typo

2. Line 140 - indicate how ChR2 was expressed

3. Figure 2 - how was illumination delivered - was it directly on the cell (radius 10 μ m) or in a larger area, for example the area in focus under the microscope?

4. Figure 2D, E - what are the dots in the bar graph? Is there a scatterplot of individual values underneath the bars? What was the laser power used for these values plotted in these panels? Please indicate number of cells in the legend.

5. Some of the text in the figures is very small, and will be hard to read in a published manuscript. Please revise.

6. Line 402 - "translaminar PV to SST connections are rare" - this is not strictly justified, as L5 PV to L2/3 SST connections were not examined.

7. Line 413 - define BC

8. Line 471 - check grammar

9. Line 473 - sentence fragment

10. Line 510 - the Munoz study examining the effects of muscarinic receptors was focused on L4 so is not relevant here

Referee #2:

The manuscript from Preuss et al. would have high impact on the field as it highlights novel inhibitory connections that build upon a substantial body of work unravelling the wS1 neuronal circuit. These novel inhibitory circuits have strong implications for understanding neuronal circuit function that have been previously demonstrated to be behaviourally relevant, within behaviourally relevant timescales (reinforced in Figure 5), may have implications for cortical synaptic plasticity, and in our understanding of how neuronal circuit function underpins sensory experience. The cortical disinhibitory circuitry (especially the VIP to SST motif) has been extensively studied for over a decade. However, most of these studies focused on disinhibitory circuits of the supragranular layers. As Preuss et al. describe, little is known about disinhibitory circuits of L5 in which SSTs cells are predominantly located. This work is original in that it clearly outlines novel connections not previously known. The experimental data is robust, and the study design cleverly uses triple-transgenic mouse lines and optogenetics to probe the neuronal circuitry. The validity of the conclusions is high. Although, the authors have not unequivocally proven that these are monosynaptic responses, but this could be easily remedied (see major comment 1, comments for the editor), which may increase the depths and understanding of these connections.

Major comments

1: Preuss et al. have taken the time to extensively study these connections, however, they have not unequivocally proven that these are monosynaptic responses. The possibility that these are polysynaptic loops has not been ruled out. Moreover, they have used ChR2 which they stimulated with a laser, as opposed to an LED (going above and beyond), that they propose spatially restricts the light within a 100um spot. Furthermore, they observe a prolonged latency in the some of the connections (VIP to L5 SST) that the authors propose may be is attributable the target location of inhibition (perisomatic versus dendritic). It is also reasonable to suspect that prolonged response latency could also be attributable to a polysynaptic response. Given that they have taken the time to be spatially specific in their ChR2 activation (which may need to be demonstrated) this question could very easily be confirmed with a 4-AP & TTX experiments, for example, or equivalent and this could greatly increase the depths and understanding of these connections.

2: Work in the supragranular layer has shown that VIP interneurons preferentially target SST interneurons. This work could imply the opposite in L5 with preferentially more and stronger connections between PV interneurons and SSTs. Can the authors consider commenting on this in the discussion as it is a novel finding with high impact?

3: In lieu of disinhibition being largely implicated with respect to impact on pyramidal neuron (PN) function/synaptic plasticity, and that the authors have demonstrated implications for frequency-dependent short-term synaptic plasticity, can the authors expound upon what this might imply downstream, perhaps in the context of facilitation versus depression?

Minor

1. For clarity is it worthwhile calling PV & VIP cells interneurons as opposed to cells. This is a very minor comment and would be fine if not taken into consideration.

2. Is it worth having a small summary circuit diagram figure summarizing how these novel circuits may fit together, perhaps in the context of what is known about the S1 microcircuit?

END OF COMMENTS

EDITOR COMMENTS

Reviewing Editor:

Dear Dr. Staiger and co-authors,

your manuscript has been reviewed by two experts in the field. As you will see both rate the quality of the recordings and reconstructions as excellent. However, the reviews also point out that the manuscript needs a major re-write to streamline the flow of information. Reviewer 1 in particular gave some some detailed examples of how to address this task. Please carefully revise your manuscript _____ accordingly.

REFEREE COMMENTS

Referee #1:

This is a very focused study providing high-quality patch-clamp recording data regarding inhibitory control of L5 somatostatin (SST) neurons. The major finding is that SST neurons are inhibited by VIP neurons regardless of layer, but by PV cells only when they reside in layer 5. The reconstructions are excellent and will contribute to databases of cellular anatomy and connectivity if made freely available. Some analysis of short term plasticity is also provided, although the significance of this measurement is unclear. One major limitation is that the diversity of L5 SST and also that of VIP neurons is not explored, and this might have an important role in establishing specific motifs for VIP and SST inhibition. Although the authors focused on SST neurons that are labeled in the GIN line, this may exclude some subtypes of SST neurons found in L5. Finally, authors indicate that these circuits are novel - this is not strictly true, as these motifs have been well-defined in multiple other studies. However, the specific details of connectivity presented here are rigorous acquired and evaluated and will be useful to experts in this subfield.

In general, the data are strong and the results convincing. However, the manuscript is written in a complex and convoluted manner that makes extracting the major conclusions difficult and detracts from the main message. This is the case throughout, from the title to the abstract and throughout the body of the text. Some recommendations about wording and vocabulary are provided below. There are also multiple tangents that detract from the main message. As it stands, the work is not ready _____ for _____ publication.

Thank you very much for recognizing the value of our data (and the study in general), despite some dissent how to describe these adequately. Since you were willing to provide very constructive criticism, we have done our best to improve the manuscript accordingly.

Main critique

1. The title is convoluted and difficult to understand.

We choose a less difficult that should be easier for the reader to understand

2. "Selectivity in translaminar disinhibitory targeting" is unnecessarily complicated. Referring to disinhibition is not really necessary in the context of this paper, since the authors are not really looking at effects on network activity of pyramidal neurons. A better focus might be the generalized finding that VIP neurons inhibit L5 SST neurons, regardless of what layer they reside in. PV neurons can serve as a contrast to this, where their inhibition is local.

We have rephrased this sentence in the Abstract to make it less complicated.

3. Throughout: "Besides" is not typically the right word. "In addition" would be better.

Changed throughout the manuscript

4. The introduction is weighed down by an unnecessary description of the vibrissae system in rodents and the 6 layers of the neocortex. This could be removed without changing the significance of the manuscript. Likewise, all the references about the "important role" that the 6-layered cortex might perform in "information processing." This is vague and also not the subject of the present study. The authors are encouraged to make reference to specific inhibitory motifs that have been proposed in other studies, and indicate how their data help revise those models.

With all due respect, we do not agree to the extent that would lead us to substantially remodel this part. For non-specialists, we think that this is a helpful maybe even necessary conceptual introduction to make it clear in which system we were performing our study and why distinguishing between intra- and translaminar circuits is important.

5. Please do not refer to "excitation-inhibition balance." This is an outdated concept of limited utility in thinking about specific networks of connectivity for controlling the activity of inhibitory neurons. Similarly, the term "disinhibitory circuitry" is unnecessarily broad and also conceptually complex - why not just talk about inhibition onto SST? Disinhibition as a topic is too expansive, because there will also be inhibition onto PV neurons, and inhibition onto VIP, two topics that are not explored in the manuscript.

Also here, we not fully agree. We are convinced that there is an E/I balance, namely one that needs continuous adjustments to behavioural requirements. We have rephrased the statement on lines X-Y to make this clear. Hopefully this will also resolve the argument to a certain degree.

We do, however, agree with the criticism toward using the term "disinhibitory circuitry". Basically, we have adopted a (bad) habit of our community. To not disconnect completely with this community, at certain places we will keep on using this "conceptual term" whereas we rephrase it where ever it is warranted.

6. References are idiosyncratic - why is Nigro 2018 cited to refer to non-L1 projecting axons? This was described significantly earlier by Agmon and colleagues. Line 68 - references about inputs to L5

neurons are similarly oddly selected or not specific to the point being made (Yao 2023?) Line 72 - why is Gouwens et al 2019 or Wu ... Fishell not cited? Line 309 - why is Pfeffer ... Scanziani or Jiang ... Tolias not cited? Line 464 - Audette ... Barth also showed this motif. Line 466 - Naskar ... Lee also rigorously examined this.

Thank you for all these reminders. Actually, as there is an ever-growing literature, sometimes the choice of one reference over the other becomes somewhat arbitrary. We adjusted our citations according to your reasonable suggestions. Just to explain one of the selections: Yao et al. 2023 did a very comprehensive study on layer- and cell type-specific connections in V1. This was considered to be a landmark study with some translational value to S1 (as it is the case with Gouwens et al. and Pfeffer et al. and so on).

7. The focus on L5a but not L5b SST neurons is unclear. Why was this done? Was the distribution of Martinotti and non-Martinotti different across L5a and 5b?

We are focussing on L5a because we were using the GIN mouse line. In this mouse line, the majority of infragranular SST cells is located in L5a, which is also discussed in the manuscript (please see l. 343-348). We partly chose the GIN line for better comparison with our previous study (Walker et al. 2016) in which the GIN mouse has also been used and for the reliability in breeding that it offered compared to other lines we would have had available. Therefore, our data has a bias towards L5a SST cells. Nevertheless, we observed all three morphological subtypes of SST cells in L5 GIN cell (T-shaped MC, fanning out MC and nMC). Therefore, we believe that the GIN mouse gives us a comprehensive and unbiased access to the whole population of L5 SST cells.

8. Data showing that L5 SST neurons have similar electrophysiological properties despite different axonal morphologies is useful. This point should be highlighted more. When reconstructions are available, the authors should show whether VIP and PV inhibition has any specificity to these two proposed subtypes. Was there any difference in rebound spikes, given a hyperpolarizing pulse, as has been suggested in some studies? Since there is some controversy about how many different subtypes of SST neurons there might be, it would be useful to the field to try to address this with the current data. Since morphological data were obtained for 65% of SST neurons targeted, the authors should at least assess the properties of different SST morphologies (fanning or T-shaped).

Thank you for pointing out this option to go into greater detail, which we also fully considered during the writing of our manuscript. However, here are the reasons why we finally decided to present the data as is (see also "Technical considerations" section of the Discussion). Some of our recorded IPSCs (especially VIP to SST) are extremely small (≤ 5 pA). We think that we are only able to see these weak connections since we are using a cesium based solution and a holding potential of 0 mV in postsynaptic SST cells. Cesium however, massively changes the parameters of analysed cells and does not allow to record physiological action potentials. Studies that observed differences in SST subtypes were mainly focussing on the active properties (action potential kinetics, afterhyperpolarization, firing pattern) (Nigro et al. 2018. Rachel et al. 2025). Due to cesium-based intracellular solution, we can only rely on the passive properties. Nevertheless, we wanted to distinguish between MC and nMC. Our passive properties data displayed that there are significant differences between MC and nMC at the population level. However, the overlap between individual MC and nMC was too large and did not allow a unambiguous separation of cell types solely based on passive properties. Cesium also does not allow us to relate connection properties to

rebound spiking. In many cases, a safe classification of fanning out or T-shaped MC was not possible without reconstruction of cells.

We also discussed whether we should further subdivide our paired recordings data according to SST cell types. We decided against this due to the low number of recovered pairs in some cases. In the L5 PV to L5 SST motif for example, only 2 cell pairs were recovered. One fanning out MC and one nMC. Therefore, we decided to add cell galleries of all recovered and reconstructed pairs. This shows the reader that we observed multiple SST subtypes in most of the connectivity motifs.

9. The number of observations (n's) is hard to find in many places. It would be easiest to put this in the figures, or if the data are not shown in figure then in the text, for each value presented.

The number of observations (n's) can now be found in all figure legends and main text. In some cases we also moved sample size numbers to more conspicuous places in the text

10. The discussion is quite elaborate and it is clear that the authors are very familiar with a large body of work. However, much of the discussion is overly speculative and not fully justified by the data presented (see line 494-6 - this is highly speculative, and the cited studies are not focused on L5 SST neurons). See also Line 517-520 - again, highly speculative and not justified by the data presented in the paper.

We have used this reminder and the suggestions therein to shorten the Discussion at appropriate places.

Minor corrections:

1. Line 115 - typo

R_i is the abbreviation of input resistance at steady state which is introduced above

2. Line 140 - indicate how Chr2 was expressed

We added the specific virus and mouse lines to the main text

3. Figure 2 - how was illumination delivered - was it directly on the cell (radius 10 μm) or in a larger area, for example the area in focus under the microscope?

We used a 100 μm diameter spot with the SST cell soma in the centre. It was already mentioned in the methods but has now been added to the main text for clarification.

4. Figure 2D, E - what are the dots in the bar graph? Is there a scatterplot of individual values underneath the bars? What was the laser power used for these values plotted in these panels? Please indicate number of cells in the legend.

We used 10-90% box and whisker plots, therefore the 0-10 % and 90-100 % intervals are shown as dots (also stated in the methods). Laser power was 10x the laser threshold to induce IPSCs, which was now added to the main text and figure legend. Number of cells was added to the figure legend.

5. Some of the text in the figures is very small, and will be hard to read in a published manuscript. Please revise.

We increased font sizes in our images.

6. Line 402 - "translaminar PV to SST connections are rare" - this is not strictly justified, as L5 PV to L2/3 SST connections were not examined.

We added L2/3 PV and L5 SST to this headline.

7. Line 413 - define BC

Defined as basket cell

8. Line 471 - check grammar

Corrected

9. Line 473 - sentence fragment

Corrected

10. Line 510 - the Munoz study examining the effects of muscarinic receptors was focused on L4 so is not relevant here

Munoz et al analysed in analysed SST cells in all layers from L2/3 to L6. Also, effects of muscarinic receptors were analysed in infragranular layer, too (see Munoz et al 2017 Fig. 4D)

Referee #2:

The manuscript from Preuss et al. would have high impact on the field as it highlights novel inhibitory connections that build upon a substantial body of work unravelling the wS1 neuronal circuit. These novel inhibitory circuits have strong implications for understanding neuronal circuit function that have been previously demonstrated to be behaviourally relevant, within behaviourally relevant timescales (reinforced in Figure 5), may have implications for cortical synaptic plasticity, and in our understanding of how neuronal circuit function underpins sensory experience. The cortical disinhibitory circuitry (especially the VIP to SST motif) has been extensively studied for over a decade. However, most of these studies focused on disinhibitory circuits of the supragranular layers. As Preuss et al. describe, little is known about disinhibitory circuits of L5 in which SSTs cells are predominantly located. This work is original in that it clearly outlines novel connections not previously known. The experimental data is robust, and the study design cleverly uses triple-transgenic mouse lines and optogenetics to probe the neuronal circuitry. The validity of the conclusions is high. Although, the authors have not unequivocally proven that these are monosynaptic responses, but this could be easily remedied (see major comment 1, comments for the editor), which may increase the depths and understanding of these connections.

Major comments

1: Preuss et al. have taken the time to extensively study these connections, however, they have not unequivocally proven that these are monosynaptic responses. The possibility that these are polysynaptic loops has not been ruled out. Moreover, they have used ChR2 which they stimulated with a laser, as opposed to an LED (going above and beyond), that they propose spatially restricts the light within a 100um spot. Furthermore, they observe a prolonged latency in the some of the connections (VIP to L5 SST) that the authors propose may be is attributable the target location of inhibition (perisomatic versus dendritic). It is also reasonable to suspect that prolonged response latency could also be attributable to a polysynaptic response. Given that they have taken the time to be spatially specific in their ChR2 activation (which may need to be demonstrated) this question could very easily be confirmed with a 4-AP & TTX experiments, for example, or equivalent and this could greatly increase the depths and understanding of these connections.

We assume that this comment is exclusively directed toward the optogenetic part of the study since in paired recordings, both mechanistically and results-wise, polysynaptic loops cannot be a confounder.

Concerning optogenetics, in our study, we injected Cre-dependent viruses into either PV-Cre or VIP-Cre animals in order to activate inhibitory cells. However, we are aware that there is a population of L5 pyramidal cells that express parvalbumin (Palicz et. al. 2024). This, population of L5 PV cells (~10 %) might also express Channelrhodopsin and therefore activate other cells. However, out latencies in ChR2 were too short and we did not observe any multi component responses. With our optogenetic experiments, we wanted to screen whether the connection is abundant enough to successfully apply paired recordings. We observed by optogenetic stimulation that L5 SST cells are strongly and abundantly innervated by PV and VIP cells. Therefore we continued with paired patch clamp recordings, where we precisely induced single or a train of action potentials in individual presynaptic inhibitory cells by direct and short (5 ms) current injection. PV, VIP and SST cells all carried fluorescent labels. We also observed some TdTomato expressing pyramidal neurons in the PV-Cre line but these cells were excluded and never patched. It is correct that VIP to SST connections

displayed prolonged latencies in comparison to PV to SST connections. However, the maximum latencies in intralaminar VIP to SST pairs was 2.614 ms and in translaminar VIP to SST pairs 2.841. These latencies are so short that it becomes highly unlikely that there was another synapse interposed. Furthermore, all VIP cells are GABAergic and thus unable to drive an interposed neuron that then would innervate the L5 SST cell.

In conclusion, we are absolutely sure that with neither method used here oligosynaptic connections were a confounder.

2: Work in the supragranular layer has shown that VIP interneurons preferentially target SST interneurons. This work could imply the opposite in L5 with preferentially more and stronger connections between PV interneurons and SSTs. Can the authors consider commenting on this in the discussion as it is a novel finding with high impact?

In our previous work, PV to SST connections were more abundant in L2/3 than VIP to SST connections (Walker et al. 2015: PV to SST 58 %, VIP to SST 35 %). This is also true in the present study (L5 PV to SST: 24.7 %, L5 VIP to L5 SST: 21.1 %, L2/3 VIP to L5 SST 29.6 %). Thus, we could not imagine how to improve the discussion on this point.

3: In lieu of disinhibition being largely implicated with respect to impact on pyramidal neuron (PN) function/synaptic plasticity, and that the authors have demonstrated implications for frequency-dependent short-term synaptic plasticity, can the authors expound upon what this might imply downstream, perhaps in the context of facilitation versus depression?

We are a little worried that fulfilling this request imposes a conflict of interest with the request of reviewer 1, so we try to be as short/concise as possible here. We added (l. 510-516) the following:

“Current hypotheses posit that disinhibition improves information processing in pyramidal cell networks (Kullander and Topolnik 2021). Given that VIP to SST cell connections show strong facilitation at high input frequencies, whereas PV to SST cell connections substantially depress at all frequencies, a high-input regime on SST cells by VIP interneurons could ensure disinhibition to be ramped up over longer time scales than those mediated by PV interneurons. Thus, disinhibition could sustain longer when VIP cells rather than PV interneurons are active as disinhibitors (Fuhrmann et al. 2002, Tremblay et al. 2016).”

Minor

1. For clarity is it worthwhile calling PV & VIP cells interneurons as opposed to cells. This is a very minor comment and would be fine if not taken into consideration.

We made the “salomonic” decision to go with both terms but more regularly with “cells” since it is more “compact” and thus reads better. It is also the preferred wording in the current literature.

2. Is it worth having a small summary circuit diagram figure summarizing how these novel circuits may

fit together, perhaps in the context of what is known about the S1 microcircuit?

We prepared a graphical abstract which contains a diagram summarizing all analysed circuits

Dear Dr Staiger,

Re: JP-RP-2025-288309R1 "Inhibitory circuit motifs of cortical somatosensory layer 5 SST neurons are uniform within layers but specific across layers" by Felix Preuss, Florian Walker, Martin Möck, Mirko Witte, and Jochen Staiger

Thank you for submitting your manuscript to The Journal of Physiology. It has been assessed by a Reviewing Editor and by 1 expert referees and we are pleased to tell you that it is acceptable for publication following satisfactory revision.

REVISION CHECKLIST:

We look forward to receiving your revised submission.

Yours sincerely,

Katalin Toth
Senior Editor
The Journal of Physiology

EDITOR COMMENTS

Senior Editor:

Please rephrase the first sentence: "In our study, we wanted to discover inhibitory circuits targeting layer 5 (L5) somatostatin- (SST) expressing neurons. " The expression 'wanted to discover' sounds a bit strange, discover could be replaced with explore, investigate etc.

Reviewing Editor:

Thank you for your appropriate revisions. Your manuscript should now be ready for publication.

REFEREE COMMENTS

Referee #2:

See attachment for comments.

END OF COMMENTS

Comments to author

This is a very thorough study using both optogenetics and whole-cell patch clamp to reveal possible novel connections between interneurons in the barrel cortex. The key findings are that L5 SST interneurons maybe widely targeted by other interneurons in the home layer with PV interneurons being the strongest connection and the reconstructions are excellent. The analysis of translaminar connections is particularly novel. It is of high importance to the field as these connections maybe key to understanding neuronal microcircuits in the sensory neocortex and could be essential in future to understanding how neuronal ensembles underpin sensory processing. The study design is well conceived, and experiments are well executed. The abstract figure is helpful and some of my comments have been addressed. My major concern is the authors have not addressed Major comment 1 from my prior review, that some of the connections may be disynaptic, this has not been tested in any experiments and/or has not been commented on. The definitive connection between these interneurons is the main messages of the manuscript and should be addressed prior to publication. Altogether this is important work to the field, it is novel, and when the comments are addressed, the work would be ready for publication.

Major comment 1:

Authors: In conclusion, we are absolutely sure that with neither method used here oligosynaptic connections were a confounder.

Response: L5 PV to L5 SST is a very clear monosynaptic response with relatively short latency and jitter (Fig 6F). A mean latency of L5 VIP to L5 SST: 324.158 ± 0.62 ms, $n = 11$; L2/3 VIP to L5 SST: 1.88 ± 0.69 ms, $n = 13$ is well within the range of a monosynaptic response, however, the jitter is relatively high, in a grey zone. Which does not rule out the possibility that these are disynaptic responses (Doyle and Andresen 2001, Sakmann 1999). The best way to rule this out would be the proposed 4-AP & TTX experiments. At the very least this should be a should be a discussion point as this could change the connectivity % (Fig 6 C) and may explain the significantly decreased synaptic success rate (Fig 6D).

Minor 1.

Authors: We made the "salomonic" decision to go with both terms but more regularly with "cells" since it is more "compact" and thus reads better. It is also the preferred wording in the current literature.

Minor 1. The authors are describing very complex neuronal circuitry and consistency is important or it becomes confusing. For example, in multiple single sentences first the interneurons are called cells, then neurons, when they are interneurons technically. Changing what one group is called in a single sentence seems to suggest different groups of groups. Then at some point neuron is not used again. Then at line 532 the term interneuron is used repeatedly. The key points then abbreviate interneurons to INs but does not use this again in the manuscript. For example:

Line 127 “To further extend our knowledge of these circuit motifs, we not only recorded from putative presynaptic PV and VIP cells in L5 but also tested for translaminal connections established by L2/3 PV and VIP neurons.”

Line 131 “We found that L5 PV, L5 VIP, and L2/3 VIP cells, but not L2/3 PV neurons, reliably target L5 SST cells with cell type-specific differences in basic synaptic properties and short-term synaptic plasticity.”

Line 220 “ Considering that we photostimulated the area around the soma of L5 SST neurons, we likely activated the axons of a population of both local and distant PV and VIP cells.”

Line 532 “ Given that VIP to SST cell connections show strong facilitation at high input frequencies, whereas PV to SST cell connections substantially depress at all frequencies, a high-input regime on SST cells by VIP interneurons could ensure disinhibition to be ramped up over longer time scales than those mediated by PV interneurons. Thus, disinhibition could sustain longer when VIP cells rather than PV interneurons are active as disinhibitors (Fuhrmann et al. 2002, Tremblay et al. 2016).”

Minor 2:

Line 571 two periods “present study..”

Comments to author

This is a very thorough study using both optogenetics and whole-cell patch clamp to reveal possible novel connections between interneurons in the barrel cortex. The key findings are that L5 SST interneurons maybe widely targeted by other interneurons in the home layer with PV interneurons being the strongest connection and the reconstructions are excellent. The analysis of translaminal connections is particularly novel. It is of high importance to the field as these connections maybe key to understanding neuronal microcircuits in the sensory neocortex and could be essential in future to understanding how neuronal ensembles underpin sensory processing. The study design is well conceived, and experiments are well executed. The abstract figure is helpful and some of my comments have been addressed. My major concern is the authors have not addressed Major comment 1 from my prior review, that some of the connections may be disynaptic, this has not been tested in any experiments and/or has not been commented on. The definitive connection between these interneurons is the main messages of the manuscript and should be addressed prior to publication. Altogether this is important work to the field, it is novel, and when the comments are addressed, the work would be ready for publication.

Many thanks for the very positive assessment of our work in general. Please find our response to the remaining concern below.

Major comment 1:

Authors: In conclusion, we are absolutely sure that with neither method used here oligosynaptic connections were a confounder.

Response: L5 PV to L5 SST is a very clear monosynaptic response with relatively short latency and jitter (Fig 6F). A mean latency of L5 VIP to L5 SST: 324.158 ± 0.62 ms, $n = 11$; L2/3 VIP to L5 SST: 1.88 ± 0.69 ms, $n = 13$ is well within the range of a monosynaptic response, however, the jitter is relatively high, in a grey zone. Which does not rule out the possibility that these are disynaptic responses (Doyle and Andresen 2001, Sakmann 1999). The best way to rule this out would be the proposed 4-AP & TTX experiments. At the very least this should be a should be a discussion point as this could change the connectivity % (Fig 6 C) and may explain the significantly decreased synaptic success rate (Fig 6D).

We still kindly disagree that the long latencies observed in VIP to SST pairs might result from disynaptic connections. We want to point out that the data points in Fig 6F show averages of individual interneuron pairs. In addition, Fig. 6B clearly shows that VIP IPSCs rise later and slower than PV inputs. Our main discussion point, which we also mentioned in the text, is the difference in synaptic location. PV cells are thought to be soma-targeting interneurons. VIP cells however, are thought to target the dendrites of their target cells. We are recording at the soma of SST interneurons. Therefore, there is a delay until synaptic currents reach the soma, especially when synapses are further away on the dendritic tree.

We are not aware of any publication in mouse cortex that shows that activation of a single presynaptic neuron leads to action potential firing in postsynaptic neurons that are at resting membrane potential. In addition, we are evoking action potential firing in a single inhibitory cell, which makes it nearly impossible to generate firing further downstream as we are hyperpolarizing these cells (if one is not assuming rebound firing). We assure, that our presynaptic cells are indeed VIP/PV IN through genetic labelling, electrophysiology (e.g. firing pattern) and often additionally through morphology. If other excitatory cells would be activated, they should also target our SST interneurons, which we would see as multicomponent responses. The fact that we only see inhibitory components and not a mixture of inhibitory and excitatory components confirms that it is an inhibitory monosynaptic response. Since we are recording IPSCs, an activated excitatory cell must activate another inhibitory neuron, which would necessitate even two serially neurons in between the recorded SST and VIP interneuron pair. The fact that our SST to VIP latencies are below 3 ms makes this impossible. As to your suggested clarifying experiment, adding 4-AP and TTX completely stops action potential firing and propagation, which is needed in our paired recordings to trigger GABA release at the synapses. Unfortunately, we are unsure which Sakmann 1999 publication is exactly meant. Doyle and Andresen 2001 however record from nucleus tractus solitaries neurons upon electrical stimulation of the solitary tract. Therefore, they activate a multitude of fibres which results in firing of an assemble of neurons. This cannot be compared to our study were we precisely activate a single inhibitory neuron.

In conclusion, we have added another sentence to our Discussion under “Technical considerations” (l. 479.483), as the alternative to doing more experiments, as kindly suggested by you.

Minor 1.

Authors: We made the “salomonic” decision to go with both terms but more regularly with “cells” since it is more “compact” and thus reads better. It is also the preferred wording in the current literature.

Minor 1. The authors are describing very complex neuronal circuitry and consistency is important or it becomes confusing. For example, in multiple single sentences first the interneurons are called cells, then neurons, when they are interneurons technically. Changing what one group is called in a single sentence seems to suggest different groups of groups. Then at some point neuron is not used again. Then at line 532 the term interneuron is used repeatedly. The key points then abbreviate interneurons to INs but does not use this again in the manuscript. For example:

Line 127 “To further extend our knowledge of these circuit motifs, we not 127 only recorded from putative presynaptic PV and VIP cells in L5 but also tested for 128 translaminar connections established by L2/3 PV and VIP neurons.”

Line 131 “We found that L5 PV, L5 VIP, and L2/3 VIP cells, but not 131 L2/3 PV neurons, reliably target L5 SST cells with cell type-specific differences in basic 132 synaptic properties and short-term synaptic plasticity.”

Line 220 “ Considering that we photostimulated the area around the soma of L5 SST neurons, 220 we likely activated the axons of a population of both local and distant PV and VIP cells.”

Line 532 “ Given that VIP to SST cell 533 connections show strong facilitation at high input frequencies, whereas PV to SST cell 534 connections substantially depress at all frequencies, a high-input regime on SST cells by VIP 535 interneurons could ensure disinhibition to be ramped up over longer time scales than those 536 mediated by PV interneurons. Thus, disinhibition could sustain longer when VIP cells rather 537 than PV interneurons are active as disinhibitors (Fuhrmann et al. 2002, Tremblay et al. 538 2016).”

We understand that this issue might become confusing, especially the examples that you presented. We therefore consistently switched to the term interneurons (IN) which should serve a better readability.

Minor 2:

Line 571 two periods “present study..”

We removed the second period

Dear Dr Staiger,

Re: JP-RP-2025-288309R2 "Inhibitory circuit motifs of cortical somatosensory layer 5 SST interneurons are uniform within layers but specific across layers" by Felix Preuss, Florian Walker, Martin Möck, Mirko Witte, and Jochen Staiger

We are pleased to tell you that your paper has been accepted for publication in The Journal of Physiology.

Yours sincerely,

Katalin Toth
Senior Editor
The Journal of Physiology

IMPORTANT POINTS TO NOTE FOLLOWING ACCEPTANCE OF YOUR PAPER:

- You can help your research get the attention it deserves! Check out Wiley's free Promotion Guide for best-practice recommendations for promoting your work at: www.wileyauthors.com/eoo/guide. You can learn more about Wiley Editing Services which offers professional video, design, and writing services to create shareable video abstracts, infographics, conference posters, lay summaries, and research news stories for your research at: www.wileyauthors.com/eoo/promotion.

- If you would like to receive our 'Research Roundup', a monthly newsletter highlighting the cutting-edge research published in The Physiological Society's family of journals (The Journal of Physiology, Experimental Physiology, Physiological Reports, The Journal of Nutritional Physiology and The Journal of Precision Medicine: Health and Disease), please click this link, fill in your name and email address and select 'Research Roundup': <https://www.physoc.org/journals-and-media/membernews>